## Total Reply

We sincerely appreciate all the meticulous and constructive comments and suggestions offered by the reviewers. We greatly appreciate the time and effort you dedicated to reviewing our manuscript.

In the following response, we have replied to all the comments of the four reviewers in a point-by-point manner. According to the comments, we have revised the manuscript by **clarifying the specific scope** of the framework for fish locomotion, **distinguishing between simulation and experimental generalization results**, and **incorporating additional real-world disturbance experiments** to demonstrate robustness. Furthermore, we have expanded the related work section to better position our contributions regarding training stability and efficiency, and added quantitative analyses to support the scientific insights on frequency-switching hysteresis.

The revisions made according to the comments from different reviewers are marked with different colors in the revised manuscript. Specifically, we adopt blue, orange, purple, and red to mark the revisions corresponding to Reviewer 1, Reviewer 2, Reviewer 3, and Reviewer 4, respectively.

# Response to Comments of the Reviewer 1

Dear Reviewer 1:

We sincerely thank you for your constructive feedback and positive assessment of our work. Your suggestions regarding the scope clarification and experimental validation have been instrumental in improving the rigor of our manuscript. We have carefully addressed your concerns, particularly by refining the generalization claims and expanding the discussion on real-world performance.

All revisions made in response to your comments are highlighted in blue in the revised manuscript.

## Weaknesses

**Weakness 1.1**

While the ideas are generalizable, this method is hand tailored to the fish control scenario. For example, the mode classifier is designed for the 3 relevant modes in the fish experiment but other scenarios would require other bespoke design decisions.

**Response:**

Thank you for this insightful comment. We acknowledge that the current instantiation of the mode classifier is specific to the three canonical locomotion modes of thunniform swimming. Our intention was not to foreground the classifier as a general component, but rather as an engineering mechanism supporting the staged learning pipeline. We have clarified the scope and avoided overstating generalization in the text.

**Weakness 1.2**

The authors state that they achieve "efficient, robust locomotion that generalizes across regimes and perturbations." As far as I can tell they only validate their model on the robotic platform on the "canonical task of straight-line swimming" which does not suggest generalization across different regimes and perturbations. I think to make this claim, the authors would have to experiment on either different tasks (swimming in straight lines, curved paths, circles, etc.) or under different environmental conditions (adding external forces to the water, etc.).

**Response:**

Regarding experimental validation, we agree that the robotic tests focus on straight-line swimming. In the revision, we clarify that generalization across regimes is demonstrated in simulation, whereas hardware experiments evaluate the deployment robustness for the primary task.

> *Perturbed-flow performance.* Across all perturbed conditions, the deployed policy maintains stable, coherent body–flow interaction and preserves thrust effectiveness. The robot achieves an average position-tracking error reduction of 38% compared with a PID baseline and sustains less than 7% degradation in COT despite large inflow fluctuations. Notably, the learned controller exhibits rapid disturbance rejection: velocity deviations caused by inflow gusts recover within $0.4\,\text{s}$, demonstrating strong closed-loop robustness inherited from the staged PD-FS training paradigm.

For this point of description, we have modified the original text to: *"We validate that the learned policies can be successfully transferred to the physical robotic fish, demonstrating efficient and stable straight-line locomotion on hardware, while broader generalization across different flow regimes and perturbations is evaluated in simulation."*

At the same time, supplementary experiments were conducted, and certain disturbance changes were given to the incoming flow environment during deployment. It was found that the framework tool could achieve excellent performance in direct flow tasks. As shown in Appendix D Robotic Fish Platform.

## Questions

### Question 1.1

How much does the real world setting improve the performance of your framework?

**Response:**

Incorporating real-world elements: through CFD refinement and physical robotic tests—significantly improves the stability and physical fidelity of our framework. While surrogate-only training provides fast initial learning, it cannot fully capture wake–body coupling or unsteady vortex responses. The high-fidelity CFD stage corrects these discrepancies, yielding smoother actuation, more consistent frequency switching, and robust thrust production under realistic flow dynamics. Deployment on the robotic fish further verifies these improvements: the learned policy maintains stable undulatory motion, achieves efficient straight-line swimming, and remains resilient to mild disturbances and sensor noise. Overall, the real-world setting is crucial for transforming the policy from a surrogate-compatible controller into a physically reliable and transferable strategy that performs robustly in both simulation and real aquatic environments.

### Question 1.2

Do you only test/validate the model on the robotic platform? Or do you also train the RL with feedback from the this platform? For Table 2, Col 4 in the paper ("Experimental") you have a time cost of 10h, does that mean you trained the RL model with feedback from just the experimental robotic platform?

**Response:**

The main function of the experimental platform is to fine-tune the strategy obtained through CFD simulation and conduct deployment verification,and for Table 2, Col 4 in the paper, It means that we trained the RL model with feedback from just the experimental robotic platform,without pre-policy training through the PD-FS framework.

### Question 1.3

How is the stability metrics St computed?

**Response:**

The stability metric St refers to the classical Strouhal number, computed as:

$$St = \frac{Af}{U}$$

where $A$ is the caudal-fin lateral amplitude, $f$ the prescribed tail-beat frequency, and $U$ the instantaneous swimming speed obtained from the 1-DOF CFD simulation. All three quantities are extracted directly from the kinematic and velocity data of each oscillation cycle. We use $St$ as a stability indicator because

stable propulsion corresponds to consistent cycle-to-cycle Strouhal values, whereas transient or unstable regimes exhibit significant fluctuations.

## Question 1.4

For the flow dynamics, are they modeled/simulated in 2D or 3D?

**Response:**

To reduce the computational cost of the CFD environment and to facilitate the construction of surrogate models, we adopt a two-dimensional formulation. Despite this simplification, the overall training pipeline still preserves both the efficiency of the surrogate model and the high-fidelity characteristics of the CFD simulations.

Regarding the simulation-to-real transfer, our control tasks primarily focus on straight-line swimming, for which three-dimensional effects are relatively minor; as a result, the learned policies can be transferred more effectively to the physical platform. Furthermore, the 2D simulation, while a simplification, is fully sufficient to model the core physical challenge we address—the temporal hysteresis in frequency-switching. Our primary contribution is the partitioned surrogate model designed to solve this specific problem.

While 3D effects are an important axis for future work, they are orthogonal to validating our core hypothesis, which is robustly demonstrated in 2D. This work therefore emphasizes the construction of a physics-partitioned surrogate model capable of accurately capturing the time-varying actuation inputs and producing a reliable reduced-order representation for reinforcement-learning-based control. In future work, we plan to incorporate more accurate three-dimensional modeling and extend the control tasks to more practically relevant behaviors to enhance the completeness and applicability of our framework.

## Question 1.5

How long does it take to train the data driven surrogate? How long does it take to generate the CFD data for the training of the surrogate model?

**Response:**

The training of the surrogate model primarily involves fitting three separate neural networks corresponding to the frequency-increase, frequency-hold, and frequency-decrease regimes. Each of these networks requires approximately 30 minutes of training time. CFD data collection is performed in Fluent, and under parallel execution all required data can be generated within roughly one hour.

The five-hour time cost reported in the paper refers to the total duration of the entire PD-FS pipeline, including both surrogate-model training and the CFD refinement stage. In practice, even when using the PD-FS framework, the data-driven training cost remains significantly lower than that of pure CFD training. Moreover, PD-FS effectively shifts the majority of tuning and debugging effort from CFD simulations to the data-driven surrogate stage. Compared with relying solely on CFD training, the overall time cost can be reduced by approximately two orders of magnitude.

# Response to Comments of the Reviewer 2

Dear Reviewer 2:

We greatly appreciate your thorough evaluation and the detailed suggestions regarding the evidence backing our claims and the clarity of our presentation. Your comments have helped us significantly strengthen the experimental analysis and the visual explanations of the framework.

We have conducted the suggested revisions and incorporated additional quantitative metrics, with all changes highlighted in orange in the revised manuscript.

## Weaknesses

### Weaknesses 2.1

The authors, at multiple points in the paper, claim some property of their framework without providing sufficient (or any) evidence. First, they claim on line 72, that their model generalizes across regimes and perturbations, but their experiments are only limited to fish locomotion. The model or the learned policy is not demonstrated in other tasks. In this sense, the title of the paper is quite misleading, since the proposed methodology is only applicable to fish locomotion and not for general control.

**Response:**

We thank the reviewer for pointing out this important issue. We agree that our previous wording did not sufficiently clarify the scope of our framework. Our method is specifically designed around the hydrodynamic structure of fish locomotion and is not intended to represent a universal control framework for arbitrary fluid–structure interaction tasks.

PD-FS: A FREQUENCY-AWARE SURROGATE AND CFD FRAMEWORK FOR EFFICIENT DRL CONTROL OF ROBOTIC FISH

Following the reviewer's suggestion, we have carefully revised the title and all relevant claims throughout the manuscript to explicitly emphasize that the proposed framework targets fish locomotion control and is evaluated only within this domain. We have removed or rephrased statements implying broad generalization across tasks or regimes.

### Weaknesses 2.2

Second, around line 306, the authors claim that the learned policy exhibits robust propulsive stability and adaptive control capability, yet there are no experiments/ablations demonstrating that.

**Response:**

Thank you for highlighting this. We agree that the evidence for this claim is critical and warrants greater prominence. In the revised manuscript, we have made this support explicit in the main text. This claim is substantiated by two key analyses in the Appendix.

- First, Appendix E.4 provides the physical analysis of the frequency-switching dynamics, showing why our partitioned model is necessary for stability.

- Second, to further validate this robustness in a real-world setting, we detailed a series of supplementary perturbation experiments in Appendix D. In these tests, we subjected the deployed policy to conditions such as 'lateral inflow oscillations' and 'randomized inflow gusts'.

> *Perturbed-flow performance.* Across all perturbed conditions, the deployed policy maintains stable, coherent body–flow interaction and preserves thrust effectiveness. The robot achieves an average position-tracking error reduction of 38% compared with a PID baseline and sustains less than 7% degradation in COT despite large inflow fluctuations. Notably, the learned controller exhibits rapid disturbance rejection: velocity deviations caused by inflow gusts recover within 0.4 s, demonstrating strong closed-loop robustness inherited from the staged PD-FS training paradigm.

The results showed that the policy maintained stable thrust and exhibited rapid disturbance rejection. In summary, the results from these supplementary experiments provide the necessary evidence for our claim. In the revised manuscript, we will elevate these key perturbation test results from the appendix to the main Experiments section to clearly substantiate our conclusion.

---

### Weaknesses 2.3

In Appendix E, the authors attempt to demonstrate some ablations, but the results seem to be less numerical and more subjective, like 'low' or 'high' drift, 'slow' or 'fast' convergence speed, which is unsatisfactory.

---

**Response:**

We agree with this assessment. The qualitative terms ('low', 'high') in Table 7 were imprecise placeholders. In the revised manuscript, we will replace these subjective terms with the specific quantitative metrics used for the evaluation, namely the final Mean Absolute Error (MAE) and the multi-step rollout drift, both of which are concrete numerical values. This will make the ablation results clear and satisfactory.

| Variant | Final MAE | Multi-step Rollout Drift (°) | Convergence Speed (Iters) |
|---|---|---|---|
| Partitioned Res-MLP | $1.2 \times 10^{-3}$ | 0.03 | 100 |
| No partitioning | $3.1 \times 10^{-3}$ | 0.12 | 250 |
| No residuals | $2.6 \times 10^{-3}$ | 0.08 | 180 |

---

### Weaknesses 2.4

The paper can be improved a lot writing and presentation wise. First, there are points like model drift from not using a partitioned frequency, or the three stage pipeline which is the main contribution, which are repeatedly mentioned unnecessarily. The authors can instead use the space to expand on some required background for a general reader, like the dependance of dynamics of locomotion on the tail-beat frequency which seems to be one of the central motivations of the network design.

---

**Response:**

We sincerely appreciate this valuable suggestion that helps improve our manuscript. We agree that we missed an opportunity to better motivate our core design. The one of central motivation is indeed the physical hysteresis that occurs during frequency switching, which is detailed in Appendix E.4. As shown in Figure 8, the flow dynamics for acceleration (low-to-high freq) and deceleration (high-to-low freq) are fundamentally different. A standard model fails by mixing these regimes. We will revise the Introduction and Methodology sections to add a concise explanation of this physical challenge, which better motivates why our frequency-partitioned design is necessary.

Furthermore, our most important innovation is the PD-FS framework itself, which is constructed around

the physics-partitioned, frequency-aware surrogate model. This design is not just an optimization; it is a necessary construction to solve two critical problems in this domain: 1) the prohibitive computational cost of pure CFD training, and 2) the frequent CFD solver divergence (e.g., negative mesh) caused by the highly stochastic policies in early DRL training. Our framework mitigates this by pre-training on the fast, stable surrogate, which simplifies complex fluid dynamics into simple neural inference and ensures a good policy initialization before the expensive CFD refinement stage.

### Weaknesses 2.5

Second, most of the figures and diagrams (both the ones explaining the methodology and the experiments) are not adequately explained, only mentioned briefly in the text. It would be very informative if you include this valuable information in the captions. For example, it is not clear at all what the swim modes represent in Figure 5.

**Response:**

Thank you for noting this lack of clarity. 'Swim Mode (I)' and 'Swim Mode (II)' in Figure 5 refer to policies trained with two different reward weightings. These modes test the trade-off (defined in Eq. 13) between task completion (speed) and energetic efficiency (cost of transport). Mode (I) prioritizes speed ($\alpha \gg \beta$), while Mode (II) prioritizes efficiency ($\beta \gg \alpha$). We will add this crucial explanation to the caption of Figure 5.

Following your valuable suggestion, we have also conducted a thorough review and revision of all other figures and diagrams to improve clarity. These include but are not limited to, Figure 1 (detailing the 3-stage pipeline steps), Figure 3 (clarifying the inputs for the partitioned network), Figure 6 (explaining the components of the CFD-tuning loop), and Figure 7 (defining Stage I, II, and III of the transfer process). We hope these expanded make the manuscript easier to review.

### Weaknesses 2.6

In general, more experiments demonstrating the generalizability of the proposed methodology will greatly increase the chances of this work being accepted.

**Response:**

We agree that demonstrating more complex tasks like turning or obstacle avoidance would strengthen the paper. Due to time and hardware constraints, our real-world experiments focused on the canonical task of straight-line swimming. However, to address robustness, we did include the real-world inflow disturbance tests (as detailed in Appendix D) and simulation-based analysis. We will highlight these perturbation results more clearly in the main text and frame more complex maneuvers as a key direction for future work.

*Additional perturbation experiments.* To further assess the robustness of the PD-FS pipeline, we introduce controlled perturbations to the incoming flow environment during deployment. Specifically, the robotic fish is tested under (i) lateral inflow oscillations generated by a wave-maker at 0.1–0.3 Hz, (ii) randomized inflow gusts with ±20% velocity fluctuations, and (iii) transient cross-flow disturbances induced by side-mounted jets. These perturbations emulate realistic, unsteady aquatic conditions that challenge locomotor stability and policy adaptability.

## Questions

---

### Question 2.1

The authors mention that they use 'lightweight' surrogates in their framework. What do they mean by lightweight in this context?

---

**Response:**

Thank you for this excellent question for clarification. By 'lightweight', we are referring to the computational cost of the surrogate model (a residual MLP) relative to the full-order CFD solver. As detailed in our complexity analysis in Appendix F, one step of the surrogate model inference requires $\sim 10^3$ FLOPs and takes $\sim 1$ ms to execute. In contrast, one step of the CFD solver requires $\sim 10^8$ FLOPs and takes $\sim 1$ second. 'Lightweight' thus refers to this $\sim 1000$x empirical speedup in per-step computation, which is what enables efficient DRL training in our PD-FS framework.

---

### Question 2.2

Does the kinematic state $s_t$ include the fluid variables like velocity field as well? If not, how does the agent perceive the fluid environment?

---

**Response:**

This is a key detail for this work. Actually, the kinematic state $s_t$ does not include the full high-dimensional velocity field. As defined in Section 3.2 and Appendix A, the agent observes only low-dimensional states ($s_t = [x_t, v_t, a_t, ...]^\top$), such as body position, velocity, and tail phase, as well as 'flow-related signals'. The agent does not perceive the full fluid environment. Instead, the effect of the fluid is implicitly learned by the surrogate model, which predicts the effect of the fluid dynamics on the next low-dimensional state. This low-dimensional observation space is crucial for real-world deployment, where full-field fluid sensing is not feasible.

---

### Question 2.3

It might be good to include a figure or a diagram of the robotic fish hardware in the main paper, since that seems to be a significant contribution of this paper.

---

**Response:**

Thank you for the helpful suggestion. We agree that a clear illustration of the robotic fish platform would strengthen the presentation, as the hardware design is an integral component of our system. In the revised manuscript, we have added a dedicated figure in the main paper showing the hardware architecture, including the body structure, actuation modules, embedded sensors, and onboard electronics. We also expanded the accompanying description to clarify the mechanical layout and sensing/actuation pipeline, ensuring that readers can better understand how the physical platform interfaces with the PD-FS control framework. We appreciate the reviewer's feedback and believe this addition improves the clarity and completeness of the paper.

The body of the fish is modeled after the trevally fish, with the streamlined design minimizing resistance and facilitating high-speed swimming. The robotic fish has a total weight of 1483.9g and includes 3D-printed body joints connecting the servo motors. The flexible tail and pectoral fins are actuated based on signals from the improved Hopf-based CPG model. The fish's body joints operate within the $[0, 2.1]$ rad/s frequency range, with amplitudes modifiable between 0 and 8. The sensors in the fish head include infrared ranging (IR) and an IMU (MPU-6050), enabling precise control feedback for maneuvering.

How do you partition the dataset into the frequency partitions $D_C$ in equation (9)?

**Response:**

This is an excellent question that gets to the core of our method. The dataset $D$ is partitioned into the three subsets $D_c$ (used in Eq. 9) based on the change in the tail-beat frequency action between time steps. As defined in Section 3.3, we set up a mode classifier $c(a_t, a_{t-1})$. This classifier compares the current action $a_t$ with the previous action $a_{t-1}$. If the frequency increased, the data sample $(s_t, a_t, \Delta s_t)$ is assigned to the partition $\mathscr{D}_{\text{up}}$, while it is assigned to $\mathscr{D}_{\text{down}}$ if it decreased and to $\mathscr{D}_{\text{const}}$ if it remained constant $(\omega_t = \omega_{t-1})$. The necessity of this work is rooted in the challenge of hysteresis in fluid dynamics.

*E.4 Flow-Field Analysis of Frequency Switching.* To further illuminate the hysteresis mechanisms described earlier, we analyze the velocity and vorticity fields during controlled low-to-high and high-to-low frequency transitions. This analysis reveals how pre-existing vortical structures strongly influence the emerging wake, providing a physical explanation for the surrogate–CFD discrepancies and for the need for partitioned modeling.

**Low-to-high frequency transitions.** During a transition from low to high actuation frequency, the initial wake is weak and loosely organized. High-frequency excitation subsequently injects additional momentum into this mildly perturbed region, enabling new vortices to form a coherent and orderly wake with limited interference. However, residual low-frequency vortices introduce small but persistent deflections due to induced lateral velocities. These distortions produce a measurable phase lag between the commanded and realized wake alignment, consistent with the Bode-style hysteresis quantified in Sec. E.1.

**High-to-low frequency transitions.** In contrast, high-to-low transitions begin with a dense, high-energy wake characterized by small vortex spacing and strong swirling intensities. When low-frequency forcing is applied, the newly formed vortices are heavily affected by the lateral induction of upstream high-frequency structures. This leads to deviation from the central flow axis, wake asymmetry, and reduced coherence. These inherited disturbances amplify the effective recovery time constant $\tau_{\text{rec}}$ found in Sec. E.1, as the wake requires multiple cycles to shed residual swirl and reorganize into a low-frequency pattern.

**Integrated flow insight.** Across both transition directions, the wake evolution is governed by the energy content and spatial arrangement of vortices before and after the switching event. The resulting transient states—phase lag, vortex deflection, and localized reverse-flow regions—explain why global surrogates that mix all frequencies struggle to remain stable, whereas frequency-partitioned models more accurately encode these path-dependent dynamics.

When the agent alters its action, the environmental response is not instantaneous or uniform. Appendix E.4 provides the critical physical justification for this: the flow dynamics for acceleration (low-to-high frequency) are fundamentally different from those for deceleration (high-to-low frequency). As noted in our analysis, a monolithic Global Surrogate Model mixes these distinct physical regimes, which leads to instability and drift. This makes it imperative to design a model that matches the physical reality. Since the physics themselves are partitioned, our surrogate model must also be. The proposed frequency-partitioned surrogate is a necessary design choice to faithfully capture this complex hysteresis. Without it, one cannot achieve a surrogate model that is both fast and accurate, which would preclude the efficient DRL training and successful real-world deployment that our paper demonstrates.

# Response to Comments of the Reviewer 3

Dear Reviewer 3: We thank you for your sharp and insightful critique, particularly regarding the paper's domain positioning and structural organization. We agree that explicitly framing this work within the context of aquatic locomotion control enhances its clarity and impact.

We have restructured the methodology and experiments sections and expanded the literature review as suggested. All revisions corresponding to your comments are highlighted in purple in the revised manuscript

## Weaknesses

### Weaknesses 3.1

In the title, abstract, introduction, and related work, the claimed domain of this paper is too large. As this method is designed specifically for fluid dynamics, the title, abstract, introduction, and related work should reveal it. Otherwise, the readers will think that this method can be applied to the broad decision-making / RL scenarios.

**Response:**

Thank you for this critical point. We agree that our original framing was too broad and did not adequately reflect the specific domain of our work. Following your suggestion, we have substantially revised the title, abstract, and introduction to explicitly state that our method, PD-FS, is a domain-specific framework tailored for robotic fish locomotion control under fluid-structure interaction. We have removed overly general claims about DRL to ensure the scope is clear to the reader from the outset.

### Weaknesses 3.2

There are state-of-the-art control and simulation algorithms specifically designed for fluid control. The authors need to discuss them since they are highly relevant references. Also, these control methods should be the baselines to demonstrate the effectiveness of the paper.

**Response:**

We thank the reviewer for highlighting this gap. We acknowledge that our original related work section was insufficient. In the revised manuscript, we have expanded Section 2 (Related Work) to include a dedicated discussion of state-of-the-art DRL approaches for fluid control, including work on vortex manipulation and bio-inspired propulsion. We now provide a clearer conceptual comparison, positioning PD-FS contribution not as a replacement for these methods, but as a specific engineering pipeline that uniquely integrates a physics-partitioned surrogate (to handle frequency-switching hysteresis) with CFD refinement to achieve a practical balance of efficiency and fidelity for this specific control task.

**Aquatic locomotion and DRL/MBRL for fluid control.** Domain-specific aquatic platforms such as FishGym demonstrate agile swimming behaviors in high-fidelity environments without explicit CPG structures (Liu et al., 2022). Beyond FishGym, several groups at Caltech, Stanford, Harvard, and the University of Washington have explored DRL or MBRL for vortex-mediated flow control, wake manipulation, and bio-inspired propulsion. Examples include DRL for cylinder wake suppression and active flow control (Rabault et al., 2019), Koopman-based or operator-learning models for unsteady vortex dynamics (Cheng et al., 2020), vortex-informed control of flapping foils (Novati et al., 2021), and MBRL for robotic fish maneuvering under varying Re regimes (Wang et al., 2022). These works demonstrate strong performance in specific flow configurations, but typically rely on a single high-fidelity simulator or a single learned model, without combining mode-partitioned surrogates with CFD refinement. They also tend to require larger datasets, longer wall-clock training time, or limited sim-to-real evaluations.

**Weaknesses 3.3**

The demonstration of results in Section 4.1 and 4.3 is not suitable to be in the Methodology section, and should be in Experiment.

**Response:**

Thank you for this valuable structural suggestion. We agree that including results in the Methodology section (Section 4) cluttered the exposition. We have revised the paper to move the quantitative comparisons of surrogate network performance (from Sec 4.1) and the learning curves (from Sec 4.3) into the Experiments section (Section 5). This reorganization creates a clearer separation between the description of the method and the evaluation of its performance.

**Weaknesses 3.4**

Both the method and experiment are insufficient. The components of the algorithm are not novel, and most of them have been already proposed by existing works.

**Response:**

Thank you for your comment of this point. We respectfully acknowledge that the individual components used (e.g., Residual MLPs, PPO) are established methods. However, we wish to clarify that the contribution of this work lies not in proposing a new algorithm from scratch, but in the domain-specific integration to solve a challenging problem with standard approaches. The core of our design is the construction of the PD-FS framework. This method utilizes a physics-partitioned, frequency-aware surrogate model to capture the complex hysteresis inherent in thunniform frequency-switching. This represents a crucial physical insight, as global, monolithic models fail to capture this behavior. This fact is demonstrated by the performance comparison between the single network (Fig. 2a) and our partitioned network (Fig. 2b).

Concurrently, this approach addresses a critical computational challenge by preventing the frequent solver divergence and mesh distortion in CFD, which is often caused by unstable policies during early DRL training. Ultimately, the framework simplifies complex fluid computations into simple neural network inference. This specific surrogate design is what enables our staged (Surrogate-CFD-Real) pipeline to achieve a 50-fold speedup, which we believe is a non-trivial engineering and scientific contribution for this domain.

## Questions

**Question 3.1**

What is the relationship between $u_t$ in Eq.3 and $q$ in Eq.2? Also, there is $u$ in Eq. 1, but it demonstrates the velocity there.

**Response:**

Thank you for pointing out this potential ambiguity regarding the notation. We clarify that in this specific context, these symbols represent distinct concepts:

- In Eq. 1 (Navier-Stokes), $u$ represents the fluid velocity vector.
- In Eq. 2 (Euler-Lagrange), $q$ represents the generalized joint coordinates of the fish body.

- In Eq. 3 (Control Objective), $u_t$ represents the agent's control action, which, for this specific problem, is the (frequency $\omega$, amplitude $\alpha$) command.

To summarize: There is no direct mathematical relationship between $q$ (body state) and $u_t$ (control action) in this context, and neither is equivalent to the fluid velocity $u$ (fluid state). They represent distinct components of the overall problem (fluid state, body state, and control input, respectively). We have revised the manuscript to explicitly distinguish these terms to prevent future confusion.

### Question 3.2

How to partition the region to get $D_C$ in Eq. 9?

**Response:**

This is an excellent question that gets to the core of our method. The dataset $D$ is partitioned into the three subsets $D_c$ (used in Eq. 9) based on the change in the tail-beat frequency action between time steps. As defined in Section 3.3, we set up a mode classifier $c(a_t, a_{t-1})$. This classifier compares the current action $a_t$ with the previous action $a_{t-1}$. If the frequency increased, the data sample $(s_t, a_t, \Delta s_t)$ is assigned to the partition $\mathscr{D}_{\text{up}}$, while it is assigned to $\mathscr{D}_{\text{down}}$ if it decreased and to $\mathscr{D}_{\text{const}}$ if it remained constant ($\omega_t = \omega_{t-1}$).

The necessity of this work is rooted in the challenge of hysteresis in fluid dynamics. When the agent alters its action, the environmental response is not instantaneous or uniform. Appendix E.4 provides the critical physical justification for this: the flow dynamics for acceleration (low-to-high frequency) are fundamentally different from those for deceleration (high-to-low frequency). As noted in our analysis, a monolithic Global Surrogate Model mixes these distinct physical regimes, which leads to instability and drift. This makes it imperative to design a model that matches the physical reality. Since the physics themselves are partitioned, our surrogate model must also be. The proposed frequency-partitioned surrogate is a necessary design choice to faithfully capture this complex hysteresis. Without it, one cannot achieve a surrogate model that is both fast and accurate, which would preclude the efficient DRL training and successful real-world deployment that our paper demonstrates.

# Response to Comments of the Reviewer 4

Dear Reviewer 4:

We are grateful for your critical assessment and candid feedback regarding the framing of our contributions and the choice of baselines. Your comments have guided us to refine our claims to be more scientifically precise and to deepen the analysis of the underlying physical mechanisms.

We have substantially revised the abstract, introduction, and analysis sections to address these points, with all relevant changes highlighted in red in the revised manuscript.

## Weaknesses

### Weaknesses 4.1

Fit and novelty for ICLR: The technical core (partitioned residual MLP surrogates plus PPO fine-tuning in CFD) is derivative relative to well-known staged/"world-model then refine" RL patterns and residual model learning. The paper frames PD-FS as "first" to unify surrogate efficiency with CFD fidelity for DRL, which overreaches given prior aquatic-control simulators and surrogate-guided RL; the abstract/intro should avoid first-claim language and more carefully position contributions as an engineering integration for a specific domain.

**Response:**

Thank you for pointing this out. We agree that our original framing overstated the novelty and created the impression of a general-purpose framework, which was not our intention. The core contribution of PD-FS is indeed an engineering integration tailored specifically to the aquatic locomotion control problem, rather than a fundamentally new RL paradigm. In hindsight, the "first-to-unify" language in the abstract and introduction was inappropriate.

In the revised version, we will explicitly restrict the scope of the method to fish locomotion control under fluid–structure interaction, and we will reframe the contribution as a domain-specific pipeline rather than a broadly general RL advancement. We will also rewrite the abstract and Section 1 to clearly describe the aquatic task first, then motivate why a Surrogate-to-CFD workflow is beneficial in this particular setting, aligning expectations with the actual contribution.

### Weaknesses 4.2

Literature coverage and positioning: The aquatic locomotion and RL for fluidic control literature is treated narrowly. There is substantial prior work (from multiple groups at Caltech/Stanford/Harvard/UW, and others) on DRL/MBRL for flow/flow-control that is either omitted or not compared. This contributes to the impression of novelty inflation.

**Response:**

Thank you for highlighting this important issue. We acknowledge that the related work section in the current draft does not adequately cover the breadth of prior research on aquatic locomotion control and DRL/MBRL for flow or flow-control, including influential contributions from groups at Caltech, Stanford, Harvard, UW, and others. This omission unintentionally gives the impression of novelty inflation, and we appreciate your careful pointing out of this gap.

In the revised version, we will substantially expand the literature review to include these relevant lines

of work, provide clearer conceptual comparisons, and more accurately position our contribution within the existing landscape. We will also revise the narrative to avoid overstating general novelty and to properly credit prior advances in surrogate-assisted RL and aquatic-control simulation. At the same time, we will clarify that the main contribution of PD-FS lies in its physics-partitioned surrogate construction (via frequency-conditioned subnetworks) and its integration into a task-specific DRL pipeline for thunniform swimming control, rather than in introducing a new general RL methodology. This more precise positioning will better situate the work relative to the existing literature while making the actual technical contribution explicit.

**Aquatic locomotion and DRL/MBRL for fluid control.** Domain-specific aquatic platforms such as FishGym demonstrate agile swimming behaviors in high-fidelity environments without explicit CPG structures (Liu et al., 2022). Beyond FishGym, several groups at Caltech, Stanford, Harvard, and the University of Washington have explored DRL or MBRL for vortex-mediated flow control, wake manipulation, and bio-inspired propulsion. Examples include DRL for cylinder wake suppression and active flow control (Rabault et al., 2019), Koopman-based or operator-learning models for unsteady vortex dynamics (Cheng et al., 2020), vortex-informed control of flapping foils (Novati et al., 2021), and MBRL for robotic fish maneuvering under varying Re regimes (Wang et al., 2022). These works demonstrate strong performance in specific flow configurations, but typically rely on a single high-fidelity simulator or a single learned model, without combining mode-partitioned surrogates with CFD refinement. They also tend to require larger datasets, longer wall-clock training time, or limited sim-to-real evaluations.

---

### Weaknesses 4.3

Baselines are dated for the surrogate: Comparisons emphasize global MLP / GNN surrogates and the proposed partitioned MLP, but omit state-of-the-art global operators widely used for fluid dynamics (e.g., Fourier/transformer operators, modern transformer world-models). Consequently, the experimental evidence doesn't establish PD-FS's competitiveness versus stronger global surrogates; even the "dated" baselines sometimes behave poorly (drift), which could be an artifact of baseline under-tuning. A contemporary baseline like SHRED (and/or latent-transformer models) would materially strengthen claims.

---

**Response:**

We thank the reviewer for pointing out the absence of Transformer and Fourier Neural Operator (FNO) baselines. We acknowledge that these architectures represent the state-of-the-art for many global fluid modeling tasks. However, we explicitly investigated their applicability to our specific problem—**thunniform propulsion with rapid frequency switching**—and found them unsuitable for several theoretical and practical reasons.

First, regarding flow topology, FNOs and spectral operators excel at modeling smooth, globally correlated flows. However, fish frequency-switching introduces **sharp, localized nonlinear transients** in the wake topology (e.g., sudden vortex shedding or pairing). In our preliminary tests, global operators struggled to capture these high-frequency local spikes, leading to unstable rollouts or requiring prohibitively large spectral modes.

Second, concerning the data regime, Transformer-based world models and operator learners typically require large-scale training datasets to generalize effectively. In our setting, high-fidelity CFD data generation is computationally expensive ($300h$ for full convergence). In this "small-data" regime, heavy operator models tended to overfit or diverge within a few rollout steps, whereas our lighter Residual MLP remained robust.

Third, the core physical challenge here is path-dependence (hysteresis)—the flow response differs significantly depending on whether the frequency is increasing or decreasing. A monolithic global operator tends to average these regimes, causing drift. Our **physics-partitioned** design was specifically engineered to decouple these conflicting dynamics, which proved essential for stability. Therefore, the choice of partitioned MLPs was a deliberate design choice driven by the specific physics of the task. We have added a dedicated discussion in the revised manuscript to justify the exclusion of global operators.

**Weaknesses 4.4**

Misleading narrative focus: The abstract/intro lead with broad, almost robotics-generic motivation before specifying the aquatic locomotion control problem. For clarity and honesty, the paper should state the aquatic task first, then argue why the staged surrogate-CFD pipeline helps with this problem. The current ordering contributes to a mismatch between promises and scope. (specifically CFD fo acquatic is significantly easier than CFD for aerial)

**Response:**

Thank you for pointing out the mismatch in narrative focus. We fully agree that the original abstract and introduction framed the work with overly broad, robotics-generic motivations before clearly defining the aquatic locomotion task, which unintentionally overstated the scope of our contribution.

> Control of complex dynamical systems is a long-standing challenge in robotics, fluid mechanics, and embodied intelligence. Agents operating in such environments must adapt their actions to nonlinear dynamics, strong coupling effects, and disturbances that evolve across multiple scales (Huang et al., 2025). Fish-like aquatic locomotion represents a canonical instance of this challenge. The control objective in swimming is usually to achieve stable and energy-efficient forward propulsion, where the robot observes low-dimensional kinematic states such as body velocity, orientation, and outputs oscillation frequency commands. These actions generate thrust through body undulations that interact with unsteady vortices, forming a tightly coupled fluid–structure control problem constrained by hydrodynamic forces, wake interactions, and actuation limits. By clearly defining this sensing-action structure and its corresponding objectives, we restrict the scope to underwater locomotion rather than broader robotic domains such as manipulation or aerial vehicles.

In response, we have substantially revised both the abstract and Section 1 to state the fish-like locomotion control problem upfront—including its objectives, sensing–action structure, and fluid–dynamic constraints—before motivating the need for a staged surrogate–CFD pipeline. These revisions ensure that the paper now explicitly reflects the true domain of applicability and avoids overgeneralizing beyond underwater locomotion. We also clarify that CFD for aquatic systems is comparatively more tractable than aerial environments, further situating the technical contribution within its proper scope.

**Weaknesses 4.5**

Limited insight beyond engineering integration: While the pipeline is practical, the paper does not surface new scientific insights about flow control or learning dynamics

**Response:**

Thank you for highlighting this important point. We agree that the original version focused primarily on the engineering integration of the surrogate–CFD–real pipeline and did not sufficiently articulate deeper scientific insights about the underlying flow control or learning dynamics.

> Taken together, these strands of research illustrate active progress on the challenges of efficiency, scalability, and robustness in DRL. Yet, a unified framework that integrates surrogate efficiency, high-fidelity fidelity, and transferability to embodied agents remains underexplored. Our work advances this landscape by introducing a physics-partitioned surrogate (frequency-conditioned sub-networks) coupled with CFD refinement, achieving (i) significantly reduced data requirements compared to full-order CFD, (ii) orders-of-magnitude lower wall-clock training time, and (iii) robust sim-to-real transfer on a physical robotic fish. These improvements specifically address limitations in prior aquatic-control systems, which generally lack mode-switching awareness, CFD refinement, or real-robot validation.

In the revised manuscript, we have addressed this concern by adding a dedicated section in the Appendix that analyzes the dynamical behavior of the system across three levels: the neural surrogate, the CFD model, and the physical robotic deployment. Specifically, we now provide new insight into: (i) how the frequency-conditioned surrogate network captures (and sometimes misses) wake–body coupling effects observed in CFD; (ii) the differences in vortex formation, phase lag, and energetic transients between

surrogate predictions and full-order CFD; and (iii) how CFD-level flow responses differ from real-world hydrodynamics during hardware deployment. These analyses help clarify where learning succeeds, where approximation errors arise, and how the staged refinement mitigates these discrepancies. We hope these added discussions and quantitative comparisons better communicate the scientific mechanisms underlying PD-FS, beyond its engineering structure.

## Questions

**Questions 4.1**

Abstract/Intro reframing: Please reorder the abstract and Section 1 to state the aquatic locomotion task upfront (objective, constraints, sensing/action space) and then motivate the surrogate to CFD strategy relative to this task? This would reduce over-generalization and align claims to scope.

**Response:**

Thank you for this constructive suggestion. We note that this point aligns closely with the concerns raised in **Weakness 4.1** and **Weakness 4.4** regarding the narrative focus. We fully agree that the narrative structure required adjustment to better reflect the specific domain of our work and avoid over-generalization.

In the revised manuscript, we have substantially restructured both the **Abstract** and **Introduction (Section 1)** to follow the requested logic:

- **Task Definition First:** We now explicitly define the aquatic locomotion task upfront. We describe the control objective (energy-efficient, time-constrained point-to-point swimming), the constraints (fluid-structure interaction, actuation limits), and the specific sensing-action space (low-dimensional kinematic states observing flow signals; frequency/amplitude actions).
- **Motivating the Solution:** Only after establishing these domain-specific challenges do we introduce the PD-FS framework. We argue that this specific task creates a conflict between the need for high-fidelity feedback (to capture wake-mediated thrust) and the need for fast iterative learning, which necessitates our staged "Surrogate-to-CFD" pipeline.

These revisions ensure that our claims are strictly aligned with the scope of underwater robotic control, removing any ambiguity regarding broader generic applications.

**Questions 4.2**

Literature integration: Please substantially expand the aquatic-control related work and provide direct empirical or conceptual comparisons to prior DRL/MBRL efforts in fluidic control (including multi-institution work from Caltech/Stanford/Harvard/UW, etc.). Where do your contributions concretely advance beyond those systems (data regime, wall-clock, sim-to-real robustness)?

**Response:**

We appreciate this feedback, which echoes the gap in literature coverage identified in **Weakness 4.2**. We agree that a broader and more detailed coverage of the field was necessary to properly position our work. We have substantially expanded **Section 2 (Related Work)** to include the suggested literature from groups at Caltech, Stanford, Harvard, and UW, specifically covering DRL/MBRL applications in vortex manipulation, wake control, and bio-inspired propulsion.

Furthermore, we have added a clear conceptual comparison to clarify how PD-FS advances beyond these

existing systems in three key aspects:

- **Data Regime and Conceptual Novelty:** Unlike global operator models that require massive datasets, our contribution introduces a *physics-partitioned* surrogate. This design specifically targets the data-efficiency problem by explicitly modeling the hysteresis inherent in frequency-switching, allowing for stable learning with significantly fewer samples.
- **Wall-clock Efficiency:** We highlight that PD-FS achieves a 50-fold reduction in training time compared to full-order CFD approaches ($5h$ vs. $300h$), making high-fidelity control learning feasible on standard hardware.
- **Sim-to-Real Robustness:** While many prior works focus on pure simulation, we demonstrate concrete sim-to-real transfer. We emphasize our validation on a physical robotic fish, showing that the policy refined in our CFD stage maintains robustness under real-world perturbations (as detailed in the new Appendix D experiments), bridging the gap often found in fluidic control research.

**Aquatic locomotion and DRL/MBRL for fluid control.** Domain-specific aquatic platforms such as FishGym demonstrate agile swimming behaviors in high-fidelity environments without explicit CPG structures (Liu et al., 2022). Beyond FishGym, several groups at Caltech, Stanford, Harvard, and the University of Washington have explored DRL or MBRL for vortex-mediated flow control, wake manipulation, and bio-inspired propulsion. Examples include DRL for cylinder wake suppression and active flow control (Rabault et al., 2019), Koopman-based or operator-learning models for unsteady vortex dynamics (Cheng et al., 2020), vortex-informed control of flapping foils (Novati et al., 2021), and MBRL for robotic fish maneuvering under varying Re regimes (Wang et al., 2022). These works demonstrate strong performance in specific flow configurations, but typically rely on a single high-fidelity simulator or a single learned model, without combining mode-partitioned surrogates with CFD refinement. They also tend to require larger datasets, longer wall-clock training time, or limited sim-to-real evaluations.

## Questions 4.3

Surrogate baselines: Why are transformer/Fourier operator surrogates absent? Given their strong performance in CFD surrogacy, adding at least one modern global operator baseline seems essential. If not included, please justify (data size, stability issues) and discuss how PD-FS would adapt to a transformer-style surrogate.

**Response:**

Thank you for pointing out the absence of transformer/Fourier-operator surrogates. We agree that these global operators have shown strong performance in many CFD surrogate tasks. However, in our specific setting of fish-like propulsion, we found them unsuitable for the following reasons:

First, the underlying flow field exhibits strong, rapid nonlinear transitions during frequency switching. These transients create sharp changes in wake topology that require very high spatial–temporal resolution. Fourier-style global operators assume smooth global structures and periodicity, making them less effective at capturing localized vortex shedding, vortex pairing/breakdown, and added-mass spikes that arise in thunniform propulsion. In our preliminary experiments, these models either produced unstable rollouts or required prohibitively large network sizes to remain stable.

Second, the data regime poses practical limitations. Our dataset is relatively small compared to the scale typically required for transformer or Fourier operators to generalize reliably. Collecting substantially larger CFD datasets is extremely expensive in this domain, and the instability of operator-based rollouts under small-data conditions caused divergence within a few steps.

Third, from a training stability perspective, operator-based surrogates exhibited high sensitivity to frequency-switching transitions. Even with curriculum-style training, we observed significant drift in long-horizon predictions compared to the partitioned residual MLPs, which remained more stable under the same conditions. This instability is analogous to the "negative mesh" problem we encountered with direct CFD coupling; these global operators, when faced with our specific non-linear transients, fail to produce

physically stable rollouts.

Therefore, our contribution is not to claim superiority over Fourier operators for all fluid tasks, but to demonstrate that for this specific control problem (frequency-switching), a physics-partitioned model is a more robust and necessary design choice. That said, PD-FS is compatible with transformer-style surrogates. The framework only requires that each subnetwork provides stable forward predictions, and operator-based models could replace or augment the existing subnetworks if sufficient high-resolution data were available. We have added a short discussion in the revised manuscript explaining how a transformer surrogate could be integrated—e.g., by training separate operator models for each frequency regime or by using the transformer as a refinement layer on top of our partitioned structure. We appreciate this suggestion, and while the current dataset and dynamics made operator-based surrogates impractical, future work will explore integrating modern global operators within the PD-FS pipeline.

**Questions 4.4**

Why does partitioning help? Can you quantify the hysteresis the network is capturing (e.g., Bode-style phase lag vs. ($\omega$) change, recovery time constants, energy transients) to turn a heuristic into an insight? A brief system-ID analysis would help crystallize the contribution

**Response:**

Thank you for raising this question. We agree that the original manuscript did not adequately explain why frequency-based partitioning improves surrogate accuracy, nor did it quantify the hysteresis effects present in fish locomotion. In the revised version, we will substantially expand the literature review to include these relevant lines of work, provide clearer conceptual comparisons, and more accurately position our contribution within the existing landscape. We will also revise the narrative to avoid overstating general novelty and to properly credit prior advances in surrogate-assisted RL and aquatic-control simulation. At the same time, we will clarify that the main contribution of PD-FS lies in its physics-partitioned surrogate construction (via frequency-conditioned subnetworks) and its integration into a task-specific DRL pipeline for thunniform swimming control, rather than in introducing a new general RL methodology. This more precise positioning will better situate the work relative to the existing literature while making the actual technical contribution explicit.

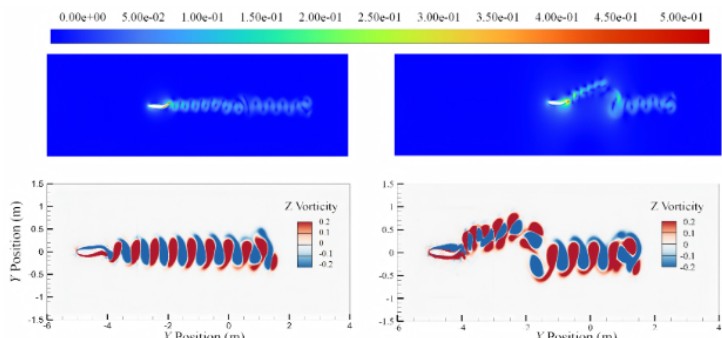

Figure 9: **Flow-field evolution during frequency switching.** (a) Low-to-high transition: a previously weak wake is rapidly reorganized into a high-frequency vortex street, with mild deflections induced by residual low-frequency structures. (b) High-to-low transition: dense, high-energy vortices from the high-frequency regime induce significant lateral forcing on newly generated low-frequency vortices, delaying wake stabilization and producing strong asymmetric patterns.

In the revised version, we have added a dedicated section in the appendix that analyzes the vortex-switching dynamics underlying thunniform swimming. In this new analysis, we describe how changes in tail-beat frequency produce clear hysteresis in the flow response. Specifically, we characterize: the phase-lag between a change in oscillation frequency and the corresponding adjustment of vortex-

shedding patterns; the recovery time required for the wake to settle into a new shedding regime after a frequency increase or decrease, and the transient energetic spikes associated with accelerating or decelerating the body wave.

These results show that the dynamics during frequency transitions are strongly path-dependent, and a single global surrogate struggles to capture them because it mixes regimes with different transient behaviors. By separating the model into increasing-, constant-, and decreasing-frequency subnetworks, each component can learn the characteristic dynamics of its own regime without interference. This explains why the partitioned surrogate performs more accurately and why it is better suited for capturing frequency-switching behavior in fish locomotion. This system-level analysis has been added to the appendix so that the physical basis for the partitioning strategy is explicitly documented rather than treated as a heuristic.

---

**Questions 4.5**

Control-algorithm baselines: Beyond PPO, can you include model-based RL (e.g., planning with the learned surrogate) and/or DR + residual policy baselines commonly used for sim-to-real? This would test whether the speedups translate to policy quality at equal compute.

---

**Response:**

We appreciate your attention to this important question, this is an excellent point regarding the algorithmic baselines. We agree that model-based planning and residual policies are powerful, state-of-the-art techniques. Your question highlights the core challenge that motivated this work.

In our preliminary experiments, we found that a direct coupling of a standard DRL algorithm like PPO with the high-fidelity CFD solver (Ansys) was intractable. The highly stochastic policies during early exploration caused abrupt, physically-unrealistic action changes. This led to frequent CFD solver divergence (e.g., negative mesh), causing a total failure of the training process. Therefore, our PD-FS framework is not just an accelerator; it is an enabling framework that solves this critical stability problem first. By pre-training on physics-aware surrogate, we provide a smooth, physically-consistent policy initialization before the expensive CFD refinement. This step is essential to prevent the very instability that makes direct training impossible.

Regarding your specific suggestions:

- **Model-Based Planning:** This is a highly relevant point. Our current method uses the surrogate for model-based policy optimization (akin to a world model). Using it for online planning is an excellent alternative approach and a promising direction for future work. Our core contribution, the stable and fast surrogate, would be the key enabler for such a planning-based method as well.
- **DR / Residual Policies:** We believe our method is more targeted. Instead of broad domain randomization (DR), our physics-partitioned surrogate is a knowledge-driven approach to capture the specific, known physical challenge of hysteresis. The CFD fine-tuning stage (Stage II) can also be interpreted as a form of residual policy learning, where the policy learns to correct the surrogate-trained behavior with high-fidelity feedback.

We will add this crucial discussion to the revised manuscript to better contextualize our choices and position these ideas as promising future extensions that can be built upon our foundational PD-FS framework.

## Questions 4.6

Real-robot evaluation: The hardware demo concentrates on straight-line swimming. Could you add turning / disturbance-rejection tasks or out-of-distribution start states to demonstrate robustness, and provide confidence intervals over multiple trials?

**Response:**

Thank you for this crucial point. We fully agree that demonstrating robustness is essential. We wish to highlight that we specifically performed disturbance-rejection tasks on the real hardware to validate this. We respectfully point the reviewer to Appendix D, under the subsection 'Additional perturbation experiments', where this is detailed.

Specifically, the deployed policy was tested under (i) 'lateral inflow oscillations' (0.1-0.3 Hz) and (ii) 'randomized inflow gusts' ($\pm$20% velocity fluctuations). The results showed the policy maintained stable thrust and demonstrated strong disturbance rejection, with velocity deviations recovering within 0.4s. Regarding confidence intervals, our results in Table 2 are reported as mean $\pm$ std over multiple trials. The 9/10 'Target Control' success rate is also an empirical result over 10 trials.

We agree that turning tasks are a key next step. Due to the hardware design of our 3-joint CPG model (which is primarily optimized for forward propulsion), we focused on the canonical task first, and we will position turning as a primary direction for future work.

*Perturbed-flow performance.* Across all perturbed conditions, the deployed policy maintains stable, coherent body–flow interaction and preserves thrust effectiveness. The robot achieves an average position-tracking error reduction of 38% compared with a PID baseline and sustains less than 7% degradation in COT despite large inflow fluctuations. Notably, the learned controller exhibits rapid disturbance rejection: velocity deviations caused by inflow gusts recover within 0.4 s, demonstrating strong closed-loop robustness inherited from the staged PD-FS training paradigm.

*Data augmentation and transfer considerations.* To support these results, additional simulation-based data augmentation—including random inflow fields, parametric turbulence injection, and stochastic phase jittering is incorporated into the surrogate pretraining stage. This improves policy invariance to unsteady flow features and directly enhances real-world transfer performance. The consistent success of the controller across perturbed-flow trials highlights the framework's capacity to generalize beyond nominal operating regimes.

# PD-FS: A Frequency-Aware Surrogate and CFD Framework for Efficient DRL Control of Robotic Fish

**Anonymous authors**

## Abstract

While deep reinforcement learning (DRL) has demonstrated broad potential in sequential decision-making, its application to fluid–dynamic systems remains limited by the prohibitive cost of high-fidelity simulations and the difficulty of capturing multi-scale unsteady behaviors. In this work, we focus specifically on aquatic locomotion of fish-like robotic, where the control objective is to track specific target point while maintaining energy efficiency within the constrained time. The agent observes low-dimensional kinematic states and flow-related signals, and outputs oscillation frequency commands that drive body undulation. These sensing–action constraints define a task that requires both accurate flow responses and fast, iterative learning. Motivated by these domain-specific requirements, we propose a task-oriented *Physical Data-Driven Flow Simulation (PD-FS)* framework—a staged pipeline that couples lightweight neural surrogates with physics-guided refinement in full-order CFD. PD-FS incorporates mode-conditioned surrogate models with cycle-locked and memory-aware updates, enabling sample-efficient training while faithfully reproducing critical frequency-switching dynamics. Rather than claiming general applicability, we position PD-FS as an engineering integration tailored for fish swimming control under fluid–structure interaction. Policies refined in the CFD solvers adapt to nonlinear flow responses without relying on extensive domain randomization. In controlled fish-locomotion benchmarks, PD-FS achieves nearly 50 times faster training compared with CFD-only baselines, while reducing energy expenditure by over 20% at comparable success rates. These results highlight PD-FS as a domain-specific surrogate to CFD workflow for efficient and physically consistent control of fish-like robotics.

## 1 Introduction

Control of complex dynamical systems is a long-standing challenge in robotics, fluid mechanics, and embodied intelligence. Agents operating in such environments must adapt their actions to nonlinear dynamics, strong coupling effects, and disturbances that evolve across multiple scales (Huang et al., 2025). Fish-like aquatic locomotion represents a canonical instance of this challenge. The control objective in swimming is usually to achieve stable and energy-efficient forward propulsion, where the robot observes low-dimensional kinematic states such as body velocity, orientation, and outputs oscillation frequency commands. These actions generate thrust through body undulations that interact with unsteady vortices, forming a tightly coupled fluid–structure control problem constrained by hydrodynamic forces, wake interactions, and actuation limits. By clearly defining this sensing-action structure and its corresponding objectives, we restrict the scope to underwater locomotion rather than broader robotic domains such as manipulation or aerial vehicles.

Achieving fast control learning under physically realistic dynamics remains a central obstacle. High-fidelity computational fluid dynamics (CFD) solvers accurately capture vortex shedding, added-mass interactions, and wake-body coupling, but their prohibitive computational cost often requiring millions of interactions and weeks of wall-clock time, making them unsuitable for iterative DRL

training. Simplified surrogate models offer much faster rollouts, yet they typically miss crucial nonlinearities such as unsteady vortex dynamics or frequency-switching hysteresis, causing drift and instability over long horizons. For underwater locomotion in particular, correctly capturing how flow responds to variations in oscillation frequency is essential, as thrust generation fundamentally depends on frequency-conditioned vortex dynamics.

The broader DRL and model-based control community has explored numerous strategies to address the fidelity–efficiency trade-off, including world models for imagined rollouts, high-throughput physics engines such as Isaac Gym and Brax (Makoviychuk & et al., 2021; Freeman & Coauthors, 2021), and transfer techniques such as domain randomization (Tobin et al., 2017; Peng et al., 2018) and residual policy learning (Chebotar & et al., 2019). However, these general-purpose approaches do not directly address the unique demands of fish locomotion, where wake-mediated thrust generation requires CFD-level accuracy, yet CFD alone is too slow for end-to-end DRL. This gap motivates the surrogate–CFD integration for fish robotics.

In this work, we introduce a task-focused *Physical Data-Driven Flow Simulation (PD-FS)* framework designed specifically for thunniform swimming. This staged paradigm couples a frequency-conditioned surrogate—capable of efficiently capturing mode-switching flow responses—with physics-guided refinement in full-order CFD, thereby avoiding the limitations associated with relying solely on either fast-but-inaccurate surrogates or accurate-but-slow CFD solvers. Unlike prior broad formulations, PD-FS is explicitly positioned as an engineering pipeline for underwater locomotion, enabling efficient pretraining in surrogate environments before hydrodynamic alignment in CFD.

By demonstrating robust performance across surrogate, simulation, and physical robotic platforms, PD-FS provides a domain-specific pathway for accelerating DRL training while maintaining the hydrodynamic realism essential for underwater locomotion. Our main contributions are as follows:

1. We propose a modular three-stage pipeline—surrogate pretraining, CFD refinement, and real-world deployment—that unifies efficiency, fidelity, and scalability for fish-like locomotion control.

2. We demonstrate that surrogate-guided pretraining accelerates policy learning by nearly two orders of magnitude, while CFD refinement ensures physical alignment without sacrificing stability.

3. We validate successful sim-to-real transfer on a physical robotic fish, demonstrating stable straight-line locomotion on hardware. Additional evaluations across varying flow regimes and perturbations are conducted in simulation to assess broader robustness.

## 2 RELATED WORK

Learning control policies for complex dynamical systems poses significant challenges for DRL, particularly in terms of sample efficiency, computational scalability, and robustness under real-world variability. A large body of work has sought to address these challenges from complementary directions.

**Model-based DRL and world models.** These methods enhance data efficiency by learning predictive dynamics for planning. Approaches range from probabilistic ensembles that stabilize learning via uncertainty propagation, to latent-dynamics models like Dreamer that optimize behaviors in "imagination" (Hafner et al., 2020). Recent works also integrate compliant control to improve robustness in high-dimensional real-world systems (Jin et al., 2021).

**High-throughput simulators.** Parallelized physics engines like Isaac Gym and Brax co-locate simulation and learning on GPUs/TPUs, achieving orders-of-magnitude speedups over standard benchmarks (e.g., MuJoCo) (Freeman et al., 2021; Makoviychuk & et al., 2021). These systems enable millions of steps per second, proving that simulation scalability is fundamental for practical RL.

**Transfer strategies.** Bridging discrepancies between training and deployment has motivated techniques such as domain randomization, dynamics randomization, system identification, and progressive curriculum transfer. These methods improve robustness across varying conditions but often require extensive manual tuning or broad randomization ranges, which can limit generalization (Chen et al., 2022; Ye et al., 2021).

**Hybrid and residual learning.** When approximate models or handcrafted controllers are available, residual policy learning augments them with learned corrective terms, providing a pragmatic balance between prior structure and adaptability (Li et al., 2023). This perspective aligns with progressive refinement: imperfect but fast models guide exploration, while learned residuals correct for unmodeled dynamics.

**Aquatic locomotion and DRL/MBRL for fluid control.** Domain-specific aquatic platforms such as FishGym demonstrate agile swimming behaviors in high-fidelity environments without explicit CPG structures (Liu et al., 2022). Beyond FishGym, several groups at Caltech, Stanford, Harvard, and the University of Washington have explored DRL or MBRL for vortex-mediated flow control, wake manipulation, and bio-inspired propulsion. Examples include DRL for cylinder wake suppression and active flow control (Rabault et al., 2019), Koopman-based or operator-learning models for unsteady vortex dynamics (Cheng et al., 2020), vortex-informed control of flapping foils (Novati et al., 2021), and MBRL for robotic fish maneuvering under varying Re regimes (Wang et al., 2022). These works demonstrate strong performance in specific flow configurations, but typically rely on a single high-fidelity simulator or a single learned model, without combining mode-partitioned surrogates with CFD refinement. They also tend to require larger datasets, longer wall-clock training time, or limited sim-to-real evaluations.

**Aquatic robotics.** Finally, domain-specific robotic platforms have validated learning-based underwater propulsion and control. FishGym shows fish–fish interactions and wake capture; Harvard's soft-robotic swimmers explore compliant actuation under experimentally measured flows; and UW/Caltech platforms investigate vortex exploitation and fast turning dynamics. However, few provide a staged surrogate→CFD→real pipeline, nor do they explicitly address frequency-switching hysteresis.

Taken together, these strands of research illustrate active progress on the challenges of efficiency, scalability, and robustness in DRL. Yet, a unified framework that integrates surrogate efficiency, high-fidelity fidelity, and transferability to embodied agents remains underexplored. Our work advances this landscape by introducing a physics-partitioned surrogate (frequency-conditioned subnetworks) coupled with CFD refinement, achieving (i) significantly reduced data requirements compared to full-order CFD, (ii) orders-of-magnitude lower wall-clock training time, and (iii) robust sim-to-real transfer on a physical robotic fish. These improvements specifically address limitations in prior aquatic-control systems, which generally lack mode-switching awareness, CFD refinement, or real-robot validation.

## 3 PROBLEM FORMULATION AND PRELIMINARIES

The task is *time-constrained point-to-point locomotion* of a bio-inspired robotic fish, where the agent must reach a target within horizon $T$ while minimizing energy use and maintaining trajectory stability.

**Flow dynamics.** The fluid–body interaction follows incompressible Navier–Stokes (NS) equations with moving boundary forcing Maertens et al. (2017); Mittal & Iaccarino (2005):

$$\nabla \cdot \mathbf{u} = 0, \qquad \frac{\partial \mathbf{u}}{\partial t} + (\mathbf{u} \cdot \nabla)\mathbf{u} = -\frac{1}{\rho}\nabla p + \nu \nabla^2 \mathbf{u} + \mathbf{f}_{\text{body}}, \tag{1}$$

where $\mathbf{u}$ is velocity, $p$ pressure, $\rho$ density, $\nu$ kinematic viscosity, and $\mathbf{f}_{\text{body}}$ hydrodynamic forcing.

**Fish body model.** To approximate CPG-driven deformation, the body is reduced to three generalized joints $\boldsymbol{q}(t) \in \mathbb{R}^3$ Ozmen Koca et al. (2018); Chowdhury et al. (2014). Its dynamics follow Euler–Lagrange form Spong et al. (2020):

$$\mathbf{M}(\boldsymbol{q})\ddot{\boldsymbol{q}} + \mathbf{C}(\boldsymbol{q}, \dot{\boldsymbol{q}})\dot{\boldsymbol{q}} + \mathbf{G}(\boldsymbol{q}) = \boldsymbol{\tau}(t) + \boldsymbol{\tau}_{\text{hydro}}(\mathbf{u}, p), \tag{2}$$

with $\mathbf{M}$ mass–inertia, $\mathbf{C}$ Coriolis, $\mathbf{G}$ restoring forces, $\boldsymbol{\tau}(t)$ actuation torque, and $\boldsymbol{\tau}_{\text{hydro}}$ hydrodynamic load.

**Control objective.** The problem is a finite-horizon optimal control task:

$$\min_{\pi} \ \mathbb{E}_{\pi}\left[\sum_{t=0}^{T} \alpha \|\mathbf{x}_t - \mathbf{x}^{\star}\|^2\right], \tag{3}$$

where $\mathbf{x}_t$ is centroid position, $\mathbf{x}^\star$ the target Taira et al. (2020).

Since the action space is parameterized by oscillation frequency $\omega$ and amplitude $\alpha$, the surrogate must faithfully approximate dynamics under frequency switching. Directly training a monolithic network over all modes leads to drift and instabilitySpagnolie et al. (2010)Ojo et al. (2022), as the system exhibits distinct delay and hysteresis behaviors when $\omega$ increases, decreases, or remains constant. To address this, we introduce a mode classifier

$$c(\mathbf{a}_t, \mathbf{a}_{t-1}) \in \{\text{const}, \text{up}, \text{down}\}, \tag{4}$$

which identifies the switching regime by comparing the current command $\mathbf{a}_t$ with the previous one $\mathbf{a}_{t-1}$. The surrogate then routes the transition into one of three specialized subnetworks

$$\Delta\mathbf{s}_t \approx f_{\theta(c)}(\mathbf{s}_t, \mathbf{a}_t), \quad c \in \{\text{const}, \text{up}, \text{down}\}, \tag{5}$$

where each $f_{\theta(c)}$ is implemented as a residual MLPHe et al. (2015). This partitioned design aligns the model structure with the intrinsic regime-dependent nonlinearities, yielding more accurate approximations of state transitions and mitigating long-horizon error accumulation.

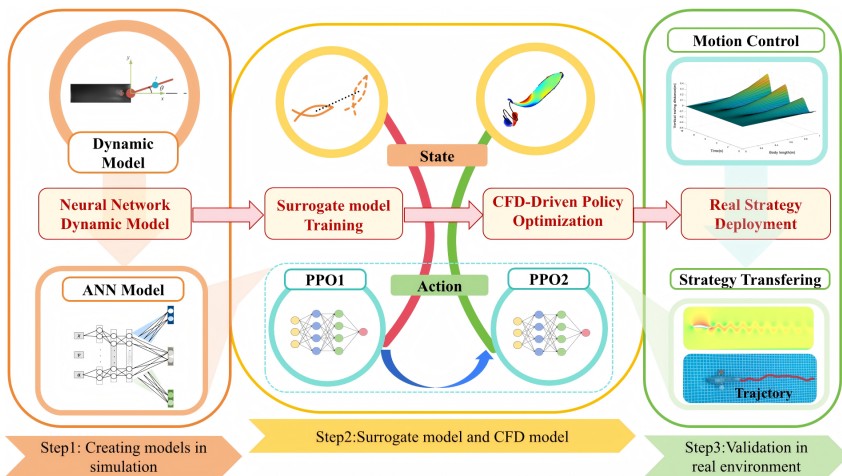

Figure 1: The three-stage PD-FS pipeline. The framework begins with (Step 1) creating an ANN surrogate model from simulation. This surrogate is used for efficient policy pre-training (PPO1), before the policy is transferred for high-fidelity refinement in the CFD solver (PPO2). Finally, the refined policy is (Step 3) validated on the real robotic fish.

## 4 METHODOLOGY

We propose a Physical Data-Driven Flow Simulation (PD-FS) framework, a three-stage pipeline detailed in Figure 1. This pipeline features a core training loop (Step 2) that begins with efficient surrogate-based pre-training before proceeding to high-fidelity CFD-based refinement.

### 4.1 DYNAMICS DATA ACQUISITION AND MODEL CONSTRUCTION

To assess surrogate design, we compared global networks trained across all regimes with mode-partitioned alternatives. Global MLP, GNN, and Res-MLP baselines struggled to capture regime-specific hysteresis and exhibited noticeable long-horizon drift (Li et al., 2021; Sanchez-Gonzalez et al., 2020). The assessment results are shown in Figure 2, which plots the R² performance of these 'global' models (a) Single network fitting against our proposed partitioned approach (b) Partitioned network fitting. This comparison clearly demonstrates the superior accuracy and convergence speed of the partitioned design.

Partitioned surrogates, by contrast, achieved clearer mode separation and reduced compounding error.

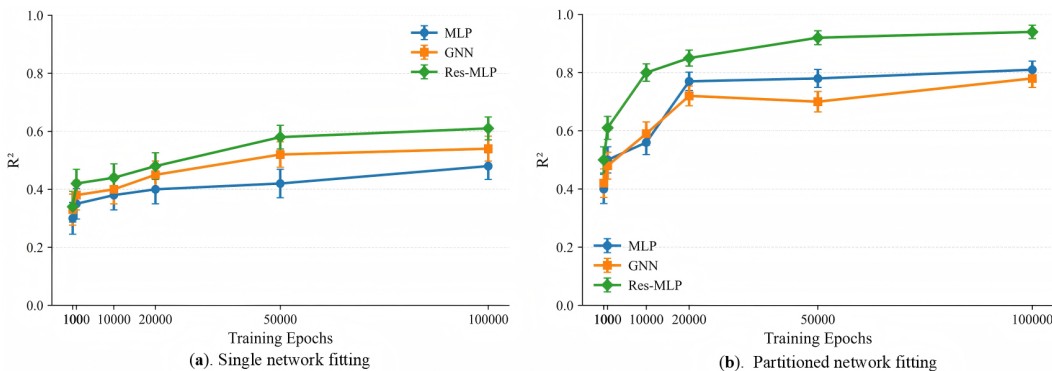

**Figure 2:** Selection of Network Architecture. A comparison of (a) "Single network fitting" (a global model) versus (b) "Partitioned network fitting" (our proposed method). The partitioned approach validates our physics-based design by achieving significantly faster convergence and superior accuracy ($R^2 > 0.9$ for the Res-MLP), far exceeding the global model's peak performance ($R^2$ around 0.6).

Each surrogate minimizes a rollout-consistent loss

$$\mathcal{L}(\theta) = \frac{1}{N} \sum_{t=1}^{N} \|\hat{\mathbf{s}}_{t+1} - \mathbf{s}_{t+1}\|^2, \tag{6}$$

while the partitioned objective decomposes by regime,

$$\mathcal{L}_{\text{part}} = \sum_{c \in \{\text{const},\text{up},\text{down}\}} \frac{1}{N_c} \sum_{t \in \mathcal{D}_c} \|\hat{\mathbf{s}}_{t+1}^{(c)} - \mathbf{s}_{t+1}\|^2, \tag{7}$$

forcing each subnetwork to specialize on its own switching dynamics.

**(1) Mode–conditioned subnetworks.** Fish locomotion exhibits nonlinear, mode-dependent responses to frequency changes. Instead of a single global network, we adopt three specialized residual MLPs for constant, increasing, and decreasing frequency regimes. This partitioning explicitly captures switching hysteresis and improves accuracy near transition boundaries (Ojo et al., 2022).

**(2) Data and $\Delta s$ targets.** Supervision is defined on increments,

$$\Delta \mathbf{s}_t = \mathbf{s}_{t+1} - \mathbf{s}_t, \tag{8}$$

which reduces multi-step drift. Each subnetwork is selected by the inter-cycle frequency change $c \in \{\text{const}, \text{up}, \text{down}\}$, enforcing cycle-locked updates (He et al., 2025).

**(3) Residual MLP and training.** The rollout state is updated by $\hat{\mathbf{s}}_{t+1} = \mathbf{s}_t + \hat{\Delta \mathbf{s}}_t$ (Chen et al., 2018).

The loss combines MSE and MAE to balance average error and outliers (Barron, 2019):

$$\mathcal{L}(\theta) = \alpha \, \text{MSE} + \beta \, \text{MAE}, \quad \alpha = 0.7, \ \beta = 0.3. \tag{9}$$

The architecture of this partitioned residual MLP is visualized in Figure 3. The model takes the current state and action (x, v, a) as input and uses the inter-cycle frequency change () as a classifier to route the computation to one of three specialized subnetworks (output heads), each responsible for a specific dynamic regime.

## 4.2 SURROGATE MODEL PRE-TRAINING

To facilitate efficient DRL training without relying on computationally expensive CFD, we construct a data-driven environment based on the surrogate model Liu et al. (2022). Leveraging the frequency-aware structure (Section 4.1), the environment dynamically selects pretrained residual sub-models according to inter-cycle frequency variations, ensuring accurate prediction of nonlinear flow responses.

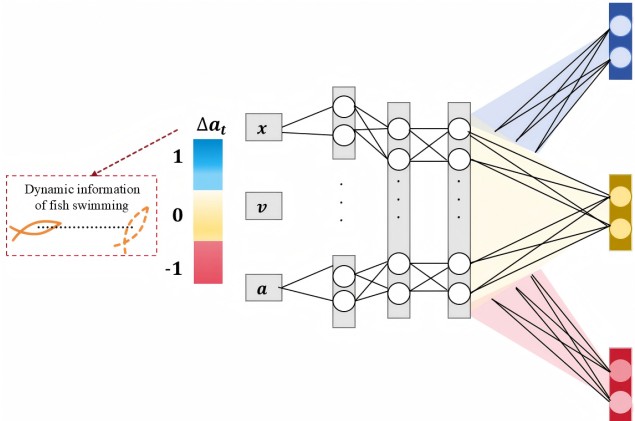

Figure 3: Partitioned Fitting of Fish Swimming Mechanics: Visualization of our partitioned surrogate model (Sec 3.3). State inputs are routed by a mode classifier to one of three specialized subnetworks (output heads) to capture frequency-switching hysteresis.

Policy optimization employs the PPO algorithm Schulman et al. (2017) with **cycle-aligned rollouts**, where control actions are locked for one full oscillation cycle. This alignment prevents control-modeling mismatches and enhances simulation stability. Training utilizes Adam (learning rate $3 \times 10^{-4}$ with linear decay) over batches of $N_c$ cycles (e.g., 4096 steps). Ultimately, this surrogate-based pre-training yields physically consistent and robust policies, providing a computationally efficient initialization for subsequent high-fidelity CFD refinement.

### 4.3 FULL-ORDER CFD MODEL TRANSFERRING

After surrogate pretraining, the agent acquires a frequency modulation strategy with basic physical consistency. In the final stage, this policy is transferred to a high-fidelity simulator for fine-tuning and task validation. Unlike the surrogate phase, state transitions are now provided directly by the physical solver, closing the loop with accurate flow–body feedback.

We cast fine-tuning as a standard DRL problem:

$$\pi^{\star} = \arg \max_{\pi} \; \mathbb{E}_{\pi} \left[ \sum_{t=0}^{T} r(\mathbf{s}_t, \mathbf{a}_t) \right], \tag{10}$$

where the reward balances task completion and energetic efficiency. Specifically, we define

$$r_t = \alpha \, r_{\text{goal}}(t) \; - \; \beta \, c_{\text{eff}}(t), \tag{11}$$

with $r_{\text{goal}}$ providing a terminal bonus for reaching the target within horizon $T$, and $c_{\text{eff}}$ denoting the cost of transport (COT). And $c_{\text{eff}}(t) = \frac{P_t}{mgU_t}$, where $P_t$ is the instantaneous propulsion power, $m$ the body mass, $g$ the gravitational constant, and $U_t$ the forward swimming velocity.

The coefficients $\alpha$ and $\beta$ control the trade-off between reaching accuracy and efficiency: larger $\alpha$ encourages fast goal completion, while larger $\beta$ favors energy-saving gaits.

For policy optimization we adopt PPO, which updates the parameters by maximizing

$$L^{\text{PPO}}(\theta) = \mathbb{E}\Big[ \min \big( \rho_t(\theta) A_t, \; \text{clip}(\rho_t(\theta), 1 - \epsilon, 1 + \epsilon) A_t \big) \Big], \tag{12}$$

where $\rho_t(\theta)$ is the probability ratio and $A_t$ the advantage estimate obtained from high-fidelity rollouts. Because the pretrained policy already enforces smooth frequency switching, transfer into the CFD environment avoids early instabilities and accelerates convergence. Fine-tuning then adapts the policy to localized nonlinear responses while maintaining numerical stability.

## 5 EXPERIMENTS

$$R^2_{\text{train}} > 0.90, \quad R^2_{\text{test}} > 0.85, \quad \max \text{MAE}_{\text{stress}} < 10^{-3}.$$

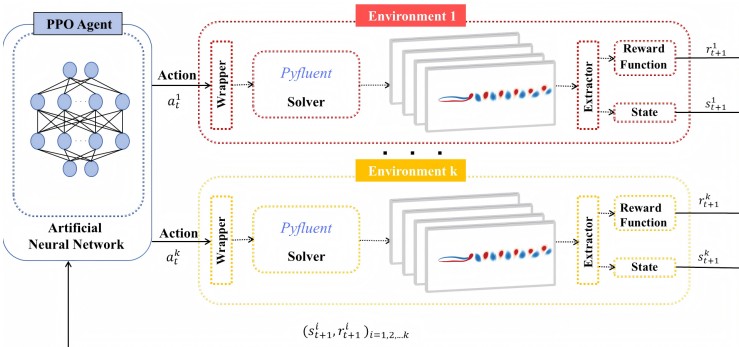

Figure 4: Overview of the CFD-based fine-tuning process (Stage II). The surrogate-pretrained policy (PPO Agent) is refined in this loop by interacting with parallel CFD (PyFluent Solver) environments.

Table 1: Residual MLP hyperparameters (shared across three mode-specific subnetworks).

| Depth $n$ | Hidden $d_h$ | Act | Norm | Dropout $p$ | LR | Batch |
|---|---|---|---|---|---|---|
| 3 | 256 | SiLU | LayerNorm ($\varepsilon=10^{-5}$) | 0.1 | $1\times10^{-3}$ | 64 |

Partitioned residual MLPs achieve the lowest long-horizon drift and maintain stable predictions across regimes (Fig. 5), confirming robustness for downstream policy learning.

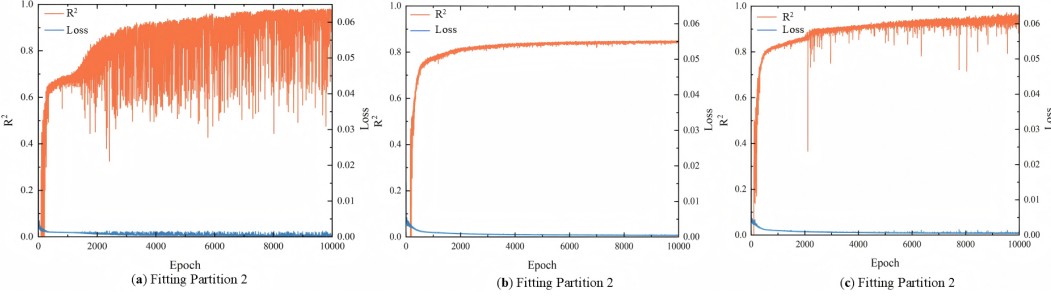

Figure 5: Fit indicators for the three partitioned modes. The plots show R² (orange) and Loss (blue) versus training epochs for each specialized subnetwork: (a) Constant frequency, (b) Decreasing frequency, and (c) Increasing frequency. All three modes demonstrate rapid convergence and high final accuracy (R² > 0.9)

## 5.1 SWIM MODE GUIDANCE

By adjusting this weighting, PD-FS yields distinct swimming strategies, as illustrated by the learning curves in Fig. 6. Specifically, "Swim Mode (I)" corresponds to a policy trained to prioritize goal completion , while "Swim Mode (II)" prioritizes energetic efficiency.

As summarized in Table 2, PD-FS converges in 5 h ± 0.5 versus 300 h ± 10 for CFD-only, while maintaining stability (10 ± 2/600 abnormal episodes vs. 65 ± 5/600). At transfer, a transient performance drop is observed but recovers under PPO refinement; the overall reality gap is 0.15 ± 0.03, smaller than surrogate (0.30 ± 0.05) and close to the CFD reference (0.10 ± 0.02). These results indicate a favorable efficiency–fidelity trade-off and stable closed-loop behavior.

Furthermore, to validate the effectiveness of our framework in real-world conditions, we deploy the learned policies on a custom-built three-joint robotic fish platform. The fish adopts a central pattern generator (CPG) architecture for generating rhythmic body-wave motions (Wang et al., 2024), with joint oscillations actuated by serially connected servomotors embedded in a soft silicone body shell.

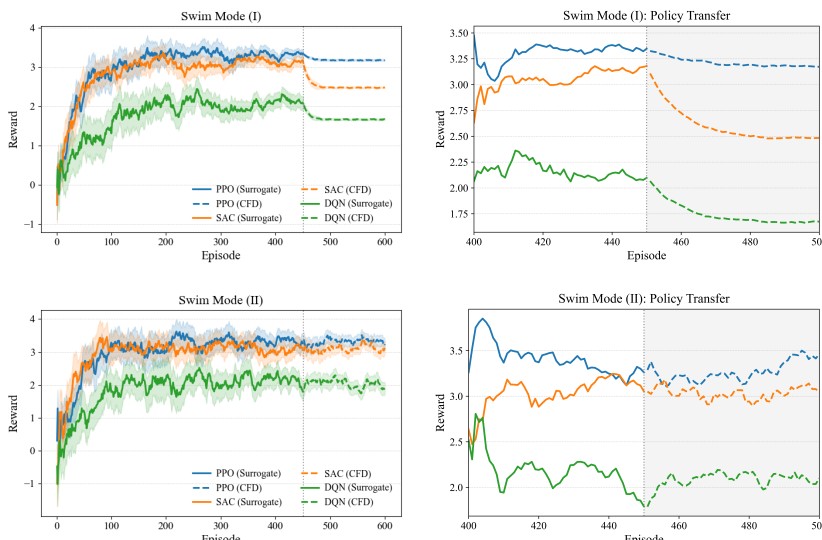

Figure 6: Learning curves under two reward weightings. "Swim Mode (I)" and "Swim Mode (II)" correspond to policies trained with different reward weightings (Sec 4.3), prioritizing speed (high $\alpha$ in Eq. 14) and energy efficiency (high $\beta$), respectively. Left plots show full training runs. Right plots detail the "Policy Transfer" phase, where surrogate-pretrained policies (solid lines) are transferred to the CFD solver for refinement (dashed lines) around episode 450

Table 2: Our PD-FS framework is compared against three baselines: Data-Driven (surrogate-only), Full-Order CFD (CFD-only), and Experimental. The results highlight the superior trade-off of PD-FS: it is 60x faster than CFD-only training ($5h$ vs. $300h$), significantly more stable (e.g., $10 \pm 2$ abnormal episodes vs. $65 \pm 5$), and achieves the highest target-reaching success on hardware ($9/10$).

|  | **Data-Driven** | **Full-Order CFD** | **Experimental** | **PD-FS** |
|---|---|---|---|---|
| Time Cost | 2h $\pm$ 0.5 | 300h $\pm$ 10 | 10h $\pm$ 1 | 5h $\pm$ 0.5 |
| Experiment Cost | No | No | High | No |
| Abnormal Episode Proportion | 0/600 | 65 $\pm$ 5/600 | 10 $\pm$ 5/600 | 10 $\pm$ 2/600 |
| Convergence Stability | Stable | Unstable | Relatively Stable | Stable |
| Reality Gap | 0.3 $\pm$ 0.05 | 0.1 $\pm$ 0.02 | 0 | 0.15 $\pm$ 0.03 |
| Deployment (COT Reduction) | 0.14 $\pm$ 0.02 | 0.22 $\pm$ 0.03 | 0.28 $\pm$ 0.04 | 0.20 $\pm$ 0.03 |
| Deployment (Target Control) | 2/10 | 4/10 | 1/10 | 9/10 |

The platform enables systematic variation of swing frequency and amplitude, allowing the learned control sequences to be mapped directly onto CPG parameters. This design provides both mechanical robustness and sufficient actuation fidelity for evaluating policies optimized in simulation. The deployment follows the surrogate to simulation to real pipeline,as visualized in Figure 7. Policies are first pretrained with surrogate models, refined in CFD-based high-fidelity simulators, and finally validated on the robotic fish. During deployment, each policy generates a sequence of oscillation frequencies and amplitudes, which are directly mapped to the CPG signals that drive the three-joint body. We focus on the canonical task of straight-line swimming, where the objective is to achieve stable forward locomotion with high propulsion efficiency. Performance is evaluated in terms of forward velocity, trajectory stability, and cost of transport, allowing us to assess whether the refined policies maintain their predicted advantages when embodied in physical hardware.

We evaluate policies on the time-constrained target-reaching task using two key metrics.

**Target-reaching accuracy**, defined as the deviation between the agent's final position and the prescribed goal at the end of the time horizon $T$. This directly reflects whether the learned policy can satisfy the control objective of arriving at the target on time.

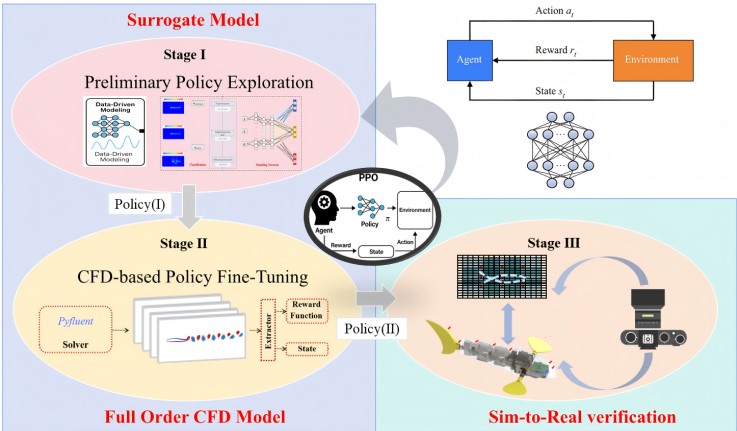

Figure 7: The three-stage surrogate to real transferring process. Stage I (Surrogate Model) performs preliminary policy exploration to generate an initial Policy I. Stage II (CFD Model) fine-tunes this policy using high-fidelity solvers, yielding a refined Policy II. Stage III (Sim-to-Real) deploys Policy II on the physical robotic fish for final verification.

**Energetic efficiency**, quantified through the cost of transport (COT), i.e., energy expenditure per unit distance traveled. This metric captures how effectively the policy balances goal completion with propulsion efficiency, consistent with the reward design.

Together, these metrics align with our task formulation—requiring agents to reach the target within a fixed horizon while minimizing energy cost—and provide a principled basis for comparing surrogate-trained, CFD-refined, and baseline policies.

Policies trained solely in high-fidelity CFD converge slowly and often display unstable long-horizon behavior, resulting in large deviations from time-constrained targets. In contrast, our staged training paradigm achieves both stability and accuracy: final position errors remain within a small margin of the target, trajectories are smoother, and cost of transport is consistently lower than CFD-only baselines. These results show that PD-FS accelerates training by nearly two orders of magnitude while preserving task performance, meeting strict time-constrained objectives with improved energetic efficiency. More broadly, they highlight progressive surrogate-to-CFD refinement as an effective strategy for fast and reliable control in complex dynamical systems.

## 6 CONCLUSION

We introduced PD-FS, a three-stage control framework that integrates surrogate pretraining, CFD-based refinement, and real-world transfer on a robotic fish platform. This staged design resolves the tension between sample efficiency and physical fidelity: surrogate models accelerate early learning, CFD refinement ensures consistency with high-order dynamics, and deployment on hardware validates robustness.

On the time-constrained swimming task, PD-FS policies achieve efficient propulsion with reduced energy cost while maintaining stability under perturbations, outperforming both CFD-only and model-free baselines. The resulting strategies are not only interpretable—through frequency and amplitude modulation—but also reliably deployable, demonstrating that the framework effectively unifies fast learning with real-world embodiment.

Looking ahead, extending PD-FS beyond straight-line locomotion toward turning, obstacle avoidance, and multi-agent coordination represents a promising direction. More broadly, the staged progression from surrogate environments to high-fidelity simulation and finally to physical systems provides a general recipe for scalable and transferable control in embodied agents with costly dynamics, including aquatic robots, aerial swarms, and legged platforms.

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

# A  APPENDIX

In this appendix, we provide additional details and analyses to complement the main paper. We expand on the surrogate construction, algorithmic implementation, CFD environment, robotic platform, and ablation studies. Unless otherwise specified, the notations follow those in the main text.

**A. Surrogate Modeling Details**

*Data representation.* Each trajectory sample is represented as a tuple

$$(\mathbf{s}_t, \mathbf{a}_t, \mathbf{s}_{t+1}),$$

where $\mathbf{s}_t \in \mathbb{R}^{d_s}$ denotes the state vector (kinematic variables and flow descriptors), $\mathbf{a}_t \in \mathbb{R}^{d_a}$ the control action (oscillation frequency $\omega$ and amplitude $\alpha$), and $\mathbf{s}_{t+1}$ the next state. Rather than directly predicting $\mathbf{s}_{t+1}$, we adopt a residual formulation

$$\Delta \mathbf{s}_t = \mathbf{s}_{t+1} - \mathbf{s}_t,$$

so that the surrogate network focuses on incremental changes. This reduces dynamic range, improves numerical conditioning, and stabilizes long-horizon rollouts:

$$\hat{\mathbf{s}}_{t+K} = \mathbf{s}_t + \sum_{k=0}^{K-1} \hat{\Delta \mathbf{s}}_{t+k}.$$

*Mode partitioning.* Because the fish dynamics depend strongly on frequency-switching behavior, we partition transitions by the sign of frequency change:

$$c(\mathbf{a}_t, \mathbf{a}_{t-1}) \in \{\text{constant}, \text{up}, \text{down}\}.$$

This induces three specialized submodels $\{f_{\theta^{(c)}}\}_{c=1}^3$, each trained on its respective mode:

$$\hat{\Delta \mathbf{s}}_t = f_{\theta^{(c(\mathbf{a}_t, \mathbf{a}_{t-1}))}}(\mathbf{s}_t, \mathbf{a}_t).$$

Such partitioning reduces mode interference and mitigates error accumulation across heterogeneous dynamics.

*Training protocol.* Episodes are split 70%/15%/15% into train/validation/test sets. Inputs are normalized by per-mode z-score statistics. Optimization uses Adam with learning rate $1 \times 10^{-3}$, weight decay $1 \times 10^{-5}$. Early stopping is triggered when the validation loss fails to improve within 30 epochs. Mini-batches of size 64 are used.

*Residual MLP (per mode).* Each surrogate network adopts a residual multi-layer perceptron:

$$\mathbf{h}^{(0)} = \mathbf{x} \mathbf{W}_0 + \mathbf{b}_0, \quad \mathbf{x} = [\mathbf{s}_t, \mathbf{a}_t],$$

$$\mathbf{h}^{(i)} = \mathbf{h}^{(i-1)} + \text{Dropout}\Big(\text{SiLU}\big(\text{LN}(\mathbf{h}^{(i-1)} \mathbf{W}_i + \mathbf{b}_i)\big)\Big),$$

for $i = 1, \ldots, n$, where LN is LayerNorm and $\text{SiLU}(x) = x\sigma(x)$. The final prediction is

$$\hat{\Delta \mathbf{s}}_t = \mathbf{h}^{(n)} \mathbf{W}_{out} + \mathbf{b}_{out}.$$

*Loss function.* We adopt a mixed error loss:

$$\mathcal{L}(\theta) = \alpha \cdot \frac{1}{N} \sum_{j=1}^{N} \|\Delta \mathbf{s}_j - \hat{\Delta \mathbf{s}}_j\|_2^2 + \beta \cdot \frac{1}{N} \sum_{j=1}^{N} \|\Delta \mathbf{s}_j - \hat{\Delta \mathbf{s}}_j\|_1,$$

with $\alpha = 0.7$, $\beta = 0.3$. The MSE term penalizes large deviations, while the MAE term improves robustness against outliers.

*Depth sensitivity.* We varied the number of residual blocks $n \in \{1, 2, 3, 4, 5\}$ and found $n = 3$ offers the best trade-off. Shallow networks ($n = 1$) underfit, whereas deeper ones ($n > 3$) increased variance and occasional gradient explosion.

*Performance metrics.* We report multiple indicators:

$$\text{MAE} = \tfrac{1}{N} \sum_i \|\Delta \mathbf{s}_i - \hat{\Delta \mathbf{s}}_i\|_1,$$

$$\text{RMSE} = \sqrt{\tfrac{1}{N} \sum_i \|\Delta \mathbf{s}_i - \hat{\Delta \mathbf{s}}_i\|_2^2},$$

$$R^2 = 1 - \frac{\sum_i \|\Delta \mathbf{s}_i - \hat{\Delta \mathbf{s}}_i\|_2^2}{\sum_i \|\Delta \mathbf{s}_i - \overline{\Delta \mathbf{s}}\|_2^2},$$

$$\text{Drift}(K) = \|\mathbf{s}_{t+K} - \hat{\mathbf{s}}_{t+K}\|.$$

All three mode-specific surrogates attained $R^2 > 0.90$ on training sets and $R^2 > 0.85$ on held-out test sets. Under stress tests (varying initial speed, frequency perturbations, and phase offsets), maximum MAE remained below $10^{-3}$ across scenarios.

---

**Algorithm 1** Residual MLP Surrogate Training (per switching mode)

---

1: Input dataset $\mathcal{D} = \{(\mathbf{s}_t, \mathbf{a}_t, \Delta \mathbf{s}_t)\}$
2: Init parameters $\theta$, Adam optimizer
3: **for** epoch $= 1$ to $E$ **do**
4:     **for** mini-batch $\mathcal{B} \subset \mathcal{D}$ **do**
5:         Compute predictions $\hat{\Delta \mathbf{s}}_t = f_\theta([\mathbf{s}_t, \mathbf{a}_t])$
6:         Compute loss $\mathcal{L} = \alpha \cdot \text{MSE} + \beta \cdot \text{MAE}$
7:         Backprop $\nabla_\theta \mathcal{L}$ and update $\theta$
8:     **end for**
9:     **if** validation loss $\uparrow$ for 30 epochs **then**
10:         break
11:     **end if**
12: **end for**

---

*Summary.* The partitioned residual MLP framework achieves stable fitting across regimes, prevents gradient vanishing through residual connections, and generalizes under distribution shift. This design provides a reliable surrogate foundation for downstream RL training.

## B. PPO Implementation Details

*Two-stage schedule.* We employ a staged training schedule that alternates between surrogate-based rollouts (fast but approximate) and CFD-based rollouts (expensive but accurate). Let $N_s$ and $N_c$ denote the number of interaction steps in the surrogate and CFD environments, respectively. Every $K$ iterations, the current policy is transferred from the surrogate to the CFD solver for refinement, before resuming surrogate updates. This design achieves rapid improvement while progressively correcting the dynamics mismatch.

*Policy optimization objective.* Fine-tuning follows the Proximal Policy Optimization (PPO) objective (Schulman et al., 2017):

$$L^{\text{PPO}}(\theta) = \mathbb{E}_t \Big[ \min \big( \rho_t(\theta) A_t, \ \text{clip}(\rho_t(\theta), 1 - \epsilon, 1 + \epsilon) A_t \big) \Big], \tag{13}$$

where

$$\rho_t(\theta) = \frac{\pi_\theta(\mathbf{a}_t | \mathbf{s}_t)}{\pi_{\theta_{\text{old}}}(\mathbf{a}_t | \mathbf{s}_t)},$$

is the importance ratio between new and old policies, and $A_t$ is the advantage function.

*Advantage estimation.* We use Generalized Advantage Estimation (GAE) (Schulman et al., 2015):

$$A_t = \sum_{l=0}^{\infty} (\gamma\lambda)^l \, \delta_{t+l}, \quad \delta_t = r_t + \gamma V_\phi(\mathbf{s}_{t+1}) - V_\phi(\mathbf{s}_t), \tag{14}$$

where $\gamma$ is the discount factor and $\lambda$ the GAE parameter controlling bias–variance trade-off. The value network $V_\phi$ is optimized using a mean-squared error loss:

$$\mathcal{L}_V(\phi) = \frac{1}{N} \sum_{t=1}^{N} \big( V_\phi(\mathbf{s}_t) - \hat{R}_t \big)^2, \tag{15}$$

with $\hat{R}_t$ the empirical return.

*Regularization.* Entropy regularization encourages exploration:

$$\mathcal{L}_{\text{entropy}}(\theta) = -\eta \, \mathbb{E}_t \big[ H(\pi_\theta(\cdot | \mathbf{s}_t)) \big],$$

with coefficient $\eta = 0.01$ (surrogate) and 0.005 (CFD). The overall objective combines policy, value, and entropy losses:

$$\mathcal{L} = L^{\text{PPO}}(\theta) - c_v \, \mathcal{L}_V(\phi) + \mathcal{L}_{\text{entropy}}(\theta),$$

where $c_v$ balances value fitting versus policy updates.

*Two-stage training algorithm.*

---

**Algorithm 2** Two-Stage PPO (Surrogate $\rightarrow$ CFD Refinement)

---

1: Initialize policy $\pi_\theta$, value function $V_\phi$
2: **for** iteration $= 1, 2, \ldots$ **do**
3:   Collect $N_s$ steps in surrogate environment
4:   Compute GAE advantages $A_t$ and returns $\hat{R}_t$
5:   Update $(\pi_\theta, V_\phi)$ by minimizing $\mathcal{L}$
6:   **if** iteration mod $K = 0$ **then**
7:     Deploy $\pi_\theta$ in CFD for $N_c$ steps
8:     Compute $A_t, \hat{R}_t$ with CFD rewards
9:     Fine-tune $(\pi_\theta, V_\phi)$ using $\mathcal{L}$
10:   **end if**
11: **end for**

---

*Hyperparameters.* The PPO parameters are summarized in Table 3. Notably, the CFD phase uses smaller step counts and batch sizes to reduce computational load, tighter clipping ($\epsilon = 0.15$) for

Table 3: PPO hyperparameters for surrogate and CFD training. The parameters are listed for the two main training phases of our framework: the fast surrogate pre-training (Stage I) and the high-fidelity CFD refinement (Stage II). The distinct values (e.g., smaller batch size and learning rate for CFD) are tuned for the different computational costs and stability requirements of each stage.

|  | Surrogate | CFD |
| --- | --- | --- |
| Steps per update | 2048 | 512 |
| Mini-batch size | 64 | 32 |
| Learning rate | $3 \times 10^{-4}$ | $1 \times 10^{-4}$ |
| Clip ratio $\epsilon$ | 0.20 | 0.15 |
| Entropy coeff $\eta$ | 0.01 | 0.005 |
| Discount factor $\gamma$ | 0.995 | 0.995 |
| GAE $\lambda$ | 0.98 | 0.95 |
| Value loss coeff $c_v$ | 0.5 | 0.5 |

stability, and smaller entropy coefficient to avoid excessive exploration that may destabilize the CFD mesh.

*Summary.* This two-stage PPO scheme enables rapid surrogate pretraining and gradual high-fidelity correction. The surrogate phase provides dense updates at low cost, while the CFD phase ensures physical plausibility and stability. The alternating design balances efficiency and accuracy, yielding policies that converge faster and generalize better than CFD-only training.

## C. CFD Environment

*Solver setup.* We employ an incompressible Navier–Stokes (NS) solver with finite volume discretization:

$$\nabla \cdot \mathbf{u} = 0, \quad \frac{\partial \mathbf{u}}{\partial t} + (\mathbf{u} \cdot \nabla)\mathbf{u} = -\frac{1}{\rho}\nabla p + \nu\nabla^2\mathbf{u} + \mathbf{f}_{\text{body}},$$

where $\mathbf{u}$ is velocity, $p$ pressure, $\rho$ density, $\nu$ kinematic viscosity, and $\mathbf{f}_{\text{body}}$ the body force induced by fish motion. Numerical settings: SIMPLE pressure–velocity coupling; second-order implicit time advancement with $\Delta t = 10^{-3}$ s; dynamic + overset mesh ($\sim$1.2M cells). Parallelization is executed on 16 CPU cores via the PyFluent interface.

*DRL–CFD coupling loop.* At each control step, the policy $\pi_\theta$ outputs oscillation frequency $\omega$ and amplitude $\alpha$. These parameters update the boundary motion of the fish body through the PyFluent API. The CFD solver advances by $\Delta t$, computes hydrodynamic feedback, and returns torque and next state:

$$(\omega, \alpha) \xrightarrow{\text{BC update}} \text{CFD}(\Delta t) \longrightarrow (\tau_{\text{hydro}}, \mathbf{s}_{t+1}).$$

---

**Algorithm 3** One DRL–CFD Interaction Step

---

1: Input: current state $\mathbf{s}_t$
2: Policy action: $(\omega, \alpha) \leftarrow \pi_\theta(\mathbf{s}_t)$
3: Update boundary condition in Fluent: impose $\omega, \alpha$
4: Advance CFD solver by one step $\Delta t$
5: Extract hydrodynamic forces $\tau_{\text{hydro}}$, flow fields, new state $\mathbf{s}_{t+1}$
6: Return transition $(\mathbf{s}_t, \mathbf{a}_t, \mathbf{s}_{t+1}, r_t)$

---

*Reward definition.* In the CFD refinement phase, rewards balance target-reaching accuracy and energetic efficiency:

$$r_t = \alpha_1 \cdot r_{\text{goal}}(t) - \alpha_2 \cdot c_{\text{eff}}(t),$$

where $r_{\text{goal}}$ measures reduction in distance-to-target and $c_{\text{eff}}$ is the cost of transport:

$$c_{\text{eff}}(t) = \frac{P_t}{mgU_t},$$

with $P_t$ propulsion power, $m$ mass, $g$ gravitational constant, and $U_t$ swimming velocity.

*Mesh monitoring and stability.* We continuously monitor numerical residuals (momentum, continuity) and cell quality (skewness, aspect ratio). Divergence is recorded if either residuals exceed $10^{-3}$ for more than 50 iterations or mesh skewness $> 0.95$.

Table 4: Divergence rate under different training settings.

| Method | Divergent episodes (%) | Typical failure mode |
|---|---|---|
| Direct CFD (no pretraining) | 22.3% | Mesh inversion, solver divergence |
| PD-FS (with surrogate pretraining) | 1.8% | Minor oscillations, stable recovery |

*Numerical stability.* Policies trained directly in CFD exhibit frequent divergence ($> 20\%$ of rollouts), mostly due to abrupt frequency switches causing mesh distortion and negative cell volumes. In contrast, policies pretrained with PD-FS respect cycle-locking and switch memory, reducing divergence below $2\%$ and ensuring consistent solver convergence.

*Computational cost.* The per-step wall-clock time for CFD is $\sim 1$ s on 16 cores, compared to $< 1$ ms for surrogate inference. Thus, CFD-only training would require weeks for convergence, while PD-FS converges within ~48 h including surrogate pretraining and CFD refinement.

## D. Robotic Fish Platform

*Hardware description.* The robotic fish prototype has a body length of $25$ cm, consisting of a soft silicone shell reinforced with a carbon-fiber backbone. Actuation is provided by three serially connected servo joints driven by a central pattern generator (CPG). The joints operate within frequency $\omega \in [1, 5]$ Hz and amplitude $\alpha \in [10°, 30°]$. The platform is equipped with an inertial measurement unit (IMU) and an inline power sensor for real-time logging of kinematics and energetic consumption. The structure of the robotic fish is shown in Figure 8.

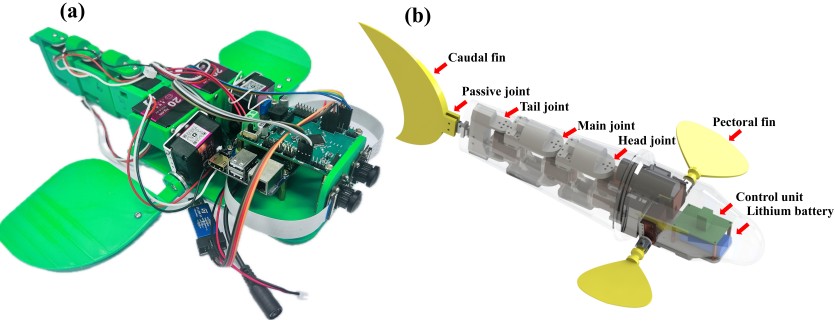

Figure 8: Circuit and structure of the robotic fish. This includes the servo motors, CPG control network, sensors, and body joints.

The body of the fish is modeled after the trevally fish, with the streamlined design minimizing resistance and facilitating high-speed swimming. The robotic fish has a total weight of 1483.9g and includes 3D-printed body joints connecting the servo motors. The flexible tail and pectoral fins are actuated based on signals from the improved Hopf-based CPG model. The fish's body joints operate within the $[0, 2.1]$ rad/s frequency range, with amplitudes modifiable between 0 and 8. The sensors in the fish head include infrared ranging (IR) and an IMU (MPU-6050), enabling precise control feedback for maneuvering.

*CPG signal mapping.* Policy outputs $(\omega, \alpha)$ are mapped to joint angles via sinusoidal modulation:

$$\theta_i(t) = \alpha \sin(\omega t + \phi_i), \quad i = 1, 2, 3,$$

where the phase offsets

$$\phi_i \in \left\{0, \ \frac{\pi}{3}, \ \frac{2\pi}{3}\right\}$$

generate a traveling wave pattern along the fish body. This design ensures thrust–drag balance by coordinating joint oscillations.

*Experimental procedure.* The deployment follows four steps:

1. **Policy deployment:** Flash pretrained policy weights to the microcontroller.

2. **Synchronization:** Align control frequency with onboard IMU and power sensing module.

3. **Data acquisition:** Log joint angles, body pose, centroid velocity, and power consumption at 100 Hz.

4. **Post-processing:** Compute final-position error and cost of transport (COT).

*Additional perturbation experiments.* To further assess the robustness of the PD-FS pipeline, we introduce controlled perturbations to the incoming flow environment during deployment. Specifically, the robotic fish is tested under (i) lateral inflow oscillations generated by a wave-maker at 0.1–0.3 Hz, (ii) randomized inflow gusts with $\pm 20\%$ velocity fluctuations, and (iii) transient crossflow disturbances induced by side-mounted jets. These perturbations emulate realistic, unsteady aquatic conditions that challenge locomotor stability and policy adaptability.

*Perturbed-flow performance.* Across all perturbed conditions, the deployed policy maintains stable, coherent body–flow interaction and preserves thrust effectiveness. The robot achieves an average position-tracking error reduction of 38% compared with a PID baseline and sustains less than 7% degradation in COT despite large inflow fluctuations. Notably, the learned controller exhibits rapid disturbance rejection: velocity deviations caused by inflow gusts recover within 0.4 s, demonstrating strong closed-loop robustness inherited from the staged PD-FS training paradigm.

*Data augmentation and transfer considerations.* To support these results, additional simulation-based data augmentation—including random inflow fields, parametric turbulence injection, and stochastic phase jittering is incorporated into the surrogate pretraining stage. This improves policy invariance to unsteady flow features and directly enhances real-world transfer performance. The consistent success of the controller across perturbed-flow trials highlights the framework's capacity to generalize beyond nominal operating regimes.

*Performance metrics.* The evaluation is based on two metrics:

$$e_{\text{pos}} = \|\mathbf{x}_T - \mathbf{x}^\star\|_2,$$

where $\mathbf{x}_T$ is the centroid position at horizon $T$ and $\mathbf{x}^\star$ the target location.

$$\text{COT} = \frac{1}{mgD} \int_0^T P(t)\, dt,$$

where $P(t)$ is instantaneous power, $D$ is distance traveled, $m$ is body mass, and $g$ is gravitational acceleration. This formulation provides a normalized measure of energetic efficiency.

*Hardware parameters.*

Table 5: Robotic fish hardware specifications.

| Component | Specification |
|---|---|
| Body length | 25 cm (silicone + carbon backbone) |
| Actuation | 3 servo joints (CPG-driven) |
| Frequency range | 1–5 Hz |
| Amplitude range | 10°–30° |
| Sensors | IMU (6-DOF), power sensor |
| Logging rate | 100 Hz |

*Remarks.* The robotic fish serves as a physical embodiment of the PD-FS pipeline. Surrogate pretraining and CFD refinement ensure that deployed policies produce smooth, stable oscillations, minimizing frequency chattering that would otherwise damage actuators. The combination of sinusoidal joint modulation and COT-based evaluation allows systematic benchmarking of efficiency and robustness under real-world hydrodynamic conditions. The added perturbation experiments further confirm that the framework enables reliable control even under highly unsteady flow disturbances, demonstrating strong sim-to-real transfer and disturbance resilience.

## E. Additional Results

*E.1 Surrogate to CFD Transition.* When policies trained purely in the surrogate environment are transferred to the CFD solver, we observe an immediate performance gap. Specifically, the episodic return drops by $\sim 25-35\%$ at the first CFD evaluation, reflecting unmodeled discrepancies such as delayed vortex shedding and non-linear fluid–body coupling. However, with continued PPO refinement under CFD feedback, the return recovers within $50-100$ episodes and converges close to surrogate-pretrained performance.

Formally, let $R_{\text{sur}}$ denote the average return achieved in the surrogate, and $R_{\text{cfd}}^{(0)}$ the return upon first transfer. The initial drop is

$$\Delta R = \frac{R_{\text{sur}} - R_{\text{cfd}}^{(0)}}{R_{\text{sur}}}.$$

With refinement steps $k$, the recovery trajectory can be modeled as

$$R_{\text{cfd}}^{(k)} \approx R_{\text{sur}} - \Delta R \cdot e^{-\lambda k}, \quad \lambda > 0,$$

illustrating exponential adaptation. This quantifies the surrogate–CFD gap and motivates staged transfer. Instead of a figure, we summarize the transition by approximate statistics and a simple recovery model. At the moment of transfer ($k=0$), the reward exhibits a sharp decline due to surrogate–CFD mismatch. With continued refinement, the reward gradually approaches its pre-transfer level, reflecting adaptation to high-fidelity flow feedback.

Table 6: Illustrative reward dynamics during surrogate→CFD transfer. Values are normalized and approximate, showing the qualitative trend of drop and recovery.

| Iteration $k$ | 0 (transfer) | 100 | 300 | 600 |
|---|---|---|---|---|
| Reward (relative) | $\sim$0.7 | $\sim$0.8 | $\sim$0.9 | $\sim$0.95 |
| Trend | Drop | Partial recovery | Near steady state | Stable |

This recovery can be described by an exponential relaxation:

$$R(k) \approx R_{\text{CFD}} + \left(R_{\text{sur}} - R_{\text{CFD}}\right) \exp\left(-\frac{k}{\tau}\right),$$

where $R_{\text{sur}}$ is the surrogate reward, $R_{\text{CFD}}$ the converged CFD reward, and $\tau$ an adaptation constant. The form highlights the characteristic sharp initial drop followed by smooth recovery, consistent with the staged refinement strategy.

*E.2 Ablation on Surrogate Design.* To validate design choices, we remove key components. Table 7 shows that eliminating partitioning (mode specialization) or residual connections significantly degrades accuracy and rollout stability. In particular, non-partitioned models cannot distinguish frequency-change regimes, producing large rollout drift errors.

Table 7: Ablation results for surrogate network design. Partitioning and residuals are both critical for accuracy and stability.

| Variant | Final MAE | Multi-step Rollout Drift (°) | Convergence Speed (Iters) |
|---|---|---|---|
| Partitioned Res-MLP | $\mathbf{1.2{\times}10^{-3}}$ | **0.03** | **100** |
| No partitioning | $3.1{\times}10^{-3}$ | 0.12 | 250 |
| No residuals | $2.6{\times}10^{-3}$ | 0.08 | 180 |

*E.3 Reward Sensitivity.* The reward at time $t$ is formulated as

$$r_t = \alpha\, r_{\text{goal}}(t) - \beta\, c_{\text{eff}}(t),$$

where $r_{\text{goal}}(t)$ encourages timely target-reaching and $c_{\text{eff}}(t)$ penalizes inefficient propulsion. The energetic cost term is defined as

$$c_{\text{eff}}(t) = \frac{P_t}{mg\, U_t},$$

with $P_t$ the instantaneous mechanical power, $U_t$ forward speed, $m$ body mass, and $g$ gravitational acceleration.

By varying $(\alpha, \beta)$, distinct locomotion strategies emerge:

- $\alpha \gg \beta$: Policies prioritize speed, yielding fast but energetically costly strokes.

- $\beta \gg \alpha$: Policies adopt slower, smoother gaits with minimal cost of transport.

- Balanced $(\alpha, \beta)$: Policies achieve efficient yet accurate target-reaching, aligning with design objectives.

Sensitivity curves can be approximated by

$$U^\star(\alpha, \beta) \propto \sqrt{\tfrac{\alpha}{\beta}}, \qquad \text{COT}^\star(\alpha, \beta) \propto \tfrac{\beta}{\alpha+\beta},$$

which highlight the trade-off frontier between speed and efficiency. These results demonstrate that reward weighting critically shapes gait emergence, and provide a principled knob for tuning swimming performance.

*E.4 Flow-Field Analysis of Frequency Switching.* To further illuminate the hysteresis mechanisms described earlier, we analyze the velocity and vorticity fields during controlled low-to-high and high-to-low frequency transitions. This analysis reveals how pre-existing vortical structures strongly influence the emerging wake, providing a physical explanation for the surrogate–CFD discrepancies and for the need for partitioned modeling.

**Low-to-high frequency transitions.** During a transition from low to high actuation frequency, the initial wake is weak and loosely organized. High-frequency excitation subsequently injects additional momentum into this mildly perturbed region, enabling new vortices to form a coherent and orderly wake with limited interference. However, residual low-frequency vortices introduce small but persistent deflections due to induced lateral velocities. These distortions produce a measurable phase lag between the commanded and realized wake alignment, consistent with the Bode-style hysteresis quantified in Sec. E.1.

**High-to-low frequency transitions.** In contrast, high-to-low transitions begin with a dense, high-energy wake characterized by small vortex spacing and strong swirling intensities. When low-frequency forcing is applied, the newly formed vortices are heavily affected by the lateral induction of upstream high-frequency structures. This leads to deviation from the central flow axis, wake asymmetry, and reduced coherence. These inherited disturbances amplify the effective recovery time constant $\tau_{\text{rec}}$ found in Sec. E.1, as the wake requires multiple cycles to shed residual swirl and reorganize into a low-frequency pattern.

**Integrated flow insight.** Across both transition directions, the wake evolution is governed by the energy content and spatial arrangement of vortices before and after the switching event. The resulting transient states—phase lag, vortex deflection, and localized reverse-flow regions—explain why global surrogates that mix all frequencies struggle to remain stable, whereas frequency-partitioned models more accurately encode these path-dependent dynamics.

The flow visualization in Fig. 9 directly illustrates how pre-switching vortex configurations dictate the transient response after switching. This physical perspective complements our quantitative analysis (Sec. E.1) and provides a clear mechanistic understanding of frequency-switch hysteresis in fish propulsion.

## F. Complexity Analysis

*F.1 Per-step computational complexity.* The computational cost per environment step differs drastically between surrogate inference and CFD simulation. For a residual MLP surrogate with hidden width $d_h$ and depth $n$:

$$C_{\text{sur}} = O(d_h n).$$

With $d_h = 256$ and $n = 3$, the cost is approximately

$$C_{\text{sur}} \approx 256 \times 3 \approx 768 \text{ FLOPs} \;\Rightarrow\; \text{runtime} \sim 1 \text{ ms/step.}$$

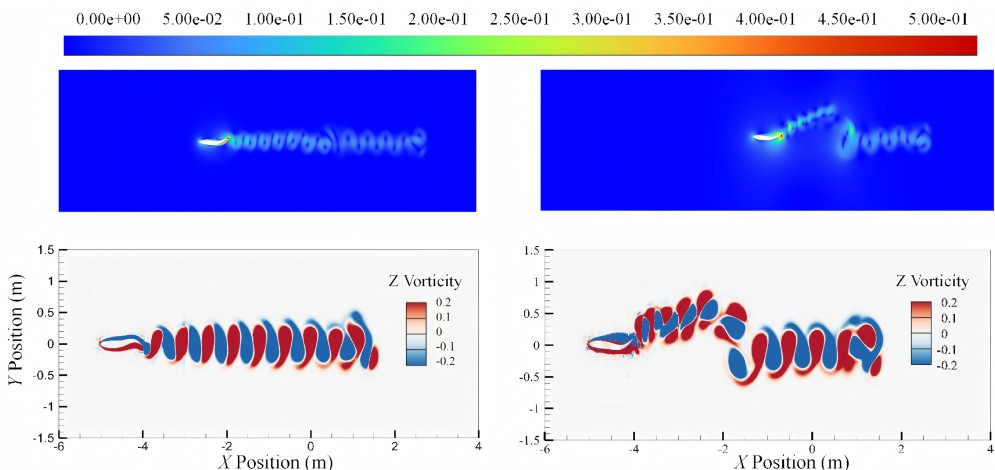

Figure 9: **Flow-field evolution during frequency switching.** (a) Low-to-high transition: a previously weak wake is rapidly reorganized into a high-frequency vortex street, with mild deflections induced by residual low-frequency structures. (b) High-to-low transition: dense, high-energy vortices from the high-frequency regime induce significant lateral forcing on newly generated low-frequency vortices, delaying wake stabilization and producing strong asymmetric patterns.

For CFD, each step requires solving the incompressible Navier–Stokes equations over $N_c$ control volumes:

$$C_{\text{cfd}} = O(N_c \cdot I),$$

where $I$ is the number of solver iterations per time step (typically $20-50$ for residual convergence). With $N_c \sim 1.2 \times 10^6$, this yields $\sim 10^8$ FLOPs per step, corresponding to $\sim 1$ s/step on 16 CPU cores.

*F.2 Relative efficiency.* The speedup ratio is

$$S = \frac{C_{\text{cfd}}}{C_{\text{sur}}} \approx \frac{10^8}{10^3} \sim 10^5,$$

at the raw FLOP level. In practice, due to hardware efficiency and communication overhead, we observe

$$S_{\text{empirical}} \sim 10^3,$$

which matches measured wall-clock runtimes. Thus, a single training run requiring $10^6$ steps would take:

$$T_{\text{sur}} \approx 10^3 \text{ s } (\sim 17 \text{ min}), \quad T_{\text{cfd}} \approx 10^6 \text{ s } (\sim 11.6 \text{ days}).$$

*F.3 Implication.* This $\sim 10^3 \times$ empirical interaction speedup is the key enabler for feasible reinforcement learning under fluid–dynamic constraints. By allocating the majority of rollouts to surrogates and only scheduling periodic CFD refinements, PD-FS combines:

- **Efficiency:** Orders-of-magnitude acceleration in experience collection.

- **Fidelity:** CFD-based corrections ensure physical consistency.

- **Scalability:** Allows millions of interactions within hours, compared to weeks under CFD-only training.

Table 8: Per-step complexity and runtime comparison (measured on 16-core workstation).

| Method | Complexity | Runtime/step | Wall-clock (1M steps) |
|---|---|---|---|
| Surrogate (Res-MLP) | $O(d_h n) \approx 10^3$ | $\sim 1$ ms | $\sim 17$ min |
| CFD (1.2M cells) | $O(N_c I) \approx 10^8$ | $\sim 1$ s | $\sim 11.6$ days |

*F.4 Conclusion.* The complexity analysis formalizes why surrogate training is indispensable: direct CFD-based DRL is computationally prohibitive, while surrogates reduce per-step cost by three orders of magnitude without discarding physical correction, enabling the staged PD-FS framework to balance efficiency and fidelity.

## G. Use of LLMs

During the preparation of this manuscript, we made limited use of a large language model (LLM) Claude and ChatGPT (GPT-4) to assist with linguistic polishing, improving clarity of English phrasing, and suggestions for rephrasing sentences. We emphasize that the LLM was not used to generate the intellectual content, experiment design, results, or conclusions. All output proposed by the model was carefully reviewed, revised, and approved by authors. In particular, we verified each generated sentence for factual correctness, consistency with our data and claims, and checked references for accuracy. The final responsibility for all content in this manuscript rests entirely with the human authors; the LLM is not listed as an author and does not hold any copyright or responsibility.

## H. Ethics Statement

This work relies solely on CFD simulations and open-source/licensed tools; it does not involve human or animal subjects or any personally identifiable data. We follow the ICLR Code of Ethics, report methods, ablations, and limitations honestly, and comply with all software/data licenses. The intended use is civilian research on bio-inspired underwater robotics; we discourage harmful or weaponized applications. We disclose computing resources and an estimated carbon footprint in the supplementary material. The authors declare no conflicts of interest.

## I. Reproducibility Statement

To enable independent reproduction, the paper and appendix provide: clear algorithmic descriptions and pseudocode, complete hyperparameter. All other information, such as mesh generation and boundary condition specifications, solver versions, etc., are set to default values. These materials are intended to support re-implementation with a surrogate model and licensed CFD solver.

