# OpenReview forum: "PD-FS:Surrogate-Enhanced Physical Data-Driven Framework for Rapid Deep Reinforcement Learning Control"
_ICLR.cc/2026/Conference — Submitted to ICLR 2026_

### Official Review · Reviewer_9kyH · 2025-10-31

**Soundness:** 2
**Presentation:** 2
**Contribution:** 1
**Rating:** 2
**Confidence:** 4

**Summary:**

The paper proposes PD-FS, a three-stage pipeline for DRL in fluid–structure interaction: (i) surrogate pre-training with a frequency-aware, mode-partitioned residual MLP; (ii) CFD refinement via PPO in a closed loop; and (iii) hardware deployment on a 3-joint robotic fish. A key modeling choice is to route state-transition prediction through three specialized subnetworks for constant / increasing / decreasing tail-beat frequencies to capture hysteresis during frequency switches. Reported outcomes include large wall-clock speedups over CFD-only training (e.g., hours vs. weeks) and improved stability during fine-tuning, with successful transfer to a robotic fish for time-constrained target reaching and reduced cost of transport.

**Strengths:**

The staged surrogate - CFD - real workflow is clearly presented, with concrete training details (partitioned residual MLPs, PPO schedules, CFD coupling) and intuitive motivation for frequency-switching gates.

The empirical story (orders-of-magnitude faster pre-training, smoother CFD fine-tuning, and lower divergence rates) is compelling for practitioners who need to reduce CFD-only training costs while maintaining fidelity in the loop.

The end-to-end demonstration of aquatic locomotion (time-constrained target reaching, cost of transport) is well-aligned with the stated control objective and shows that the method can be embodied on a physical platform.

**Weaknesses:**

(A) *Fit and novelty for ICLR*: The technical core (partitioned residual MLP surrogates plus PPO fine-tuning in CFD) is derivative relative to well-known staged/“world-model then refine” RL patterns and residual model learning. The paper frames PD-FS as “first” to unify surrogate efficiency with CFD fidelity for DRL, which overreaches given prior aquatic-control simulators and surrogate-guided RL; the abstract/intro should avoid first-claim language and more carefully position contributions as an engineering integration for a specific domain.

(B) *Literature coverage and positioning*: The aquatic locomotion and RL for fluidic control literature is treated narrowly. There is substantial prior work (from multiple groups at Caltech/Stanford/Harvard/UW, and others) on DRL/MBRL for flow/flow-control that is either omitted or not compared. This contributes to the impression of novelty inflation.

(C) *Baselines are dated for the surrogate*: Comparisons emphasize global MLP / GNN surrogates and the proposed partitioned MLP, but omit state-of-the-art global operators widely used for fluid dynamics (e.g., Fourier/transformer operators, modern transformer world-models). Consequently, the experimental evidence doesn’t establish PD-FS’s competitiveness versus stronger global surrogates; even the “dated” baselines sometimes behave poorly (drift), which could be an artifact of baseline under-tuning. A contemporary baseline like SHRED (and/or latent-transformer models) would materially strengthen claims.

(D) *Misleading narrative focus*: The abstract/intro lead with broad, almost robotics-generic motivation before specifying the aquatic locomotion control problem. For clarity and honesty, the paper should state the aquatic task first, then argue why the staged surrogate-CFD pipeline helps with this problem. The current ordering contributes to a mismatch between promises and scope. (specifically CFD fo acquatic is **significantly** easier than CFD for aerial)

(E) *Limited insight beyond engineering integration*: While the pipeline is practical, the paper does not surface new scientific insights about flow control or learning dynamics

**Questions:**

Abstract/Intro reframing: Please reorder the abstract and Section 1 to state the aquatic locomotion task upfront (objective, constraints, sensing/action space) and then motivate the surrogate to CFD strategy relative to this task? This would reduce over-generalization and align claims to scope.

Literature integration: Please substantially expand the aquatic-control related work and provide direct empirical or conceptual comparisons to prior DRL/MBRL efforts in fluidic control (including multi-institution work from Caltech/Stanford/Harvard/UW, etc.). Where do your contributions concretely advance beyond those systems (data regime, wall-clock, sim-to-real robustness)?

Surrogate baselines: Why are transformer/Fourier operator surrogates absent? Given their strong performance in CFD surrogacy, adding at least one modern global operator baseline seems essential. If not included, please justify (data size, stability issues) and discuss how PD-FS would adapt to a transformer-style surrogate.

Why does partitioning help? Can you quantify the hysteresis the network is capturing (e.g., Bode-style phase lag vs. (\omega) change, recovery time constants, energy transients) to turn a heuristic into an insight? A brief system-ID analysis would help crystallize the contribution

Control-algorithm baselines: Beyond PPO, can you include model-based RL (e.g., planning with the learned surrogate) and/or DR + residual policy baselines commonly used for sim-to-real? This would test whether the speedups translate to policy quality at equal compute.

Real-robot evaluation: The hardware demo concentrates on straight-line swimming. Could you add turning / disturbance-rejection tasks or out-of-distribution start states to demonstrate robustness, and provide confidence intervals over multiple trials?

---

### Official Review · Reviewer_WxPG · 2025-10-31

**Soundness:** 2
**Presentation:** 2
**Contribution:** 2
**Rating:** 2
**Confidence:** 4

**Summary:**

The paper proposes a Physical Data-Driven Flow Simulation (PD-FS) framework, which incorporates mode-conditioned surrogate models, enabling sample-efficient training while faithfully reproducing critical frequency-switching dynamics. Then, policies refined in CFD solvers adapt to nonlinear flow responses. In experiments, PD-FS achieves faster training compared with CFD baselines, while reducing energy expenditure at comparable success rates.

**Strengths:**

- The paper proposes mode-conditioned surrogate models with cycle-locked and memory-aware updates.
- Policies do not need domain randomization.

**Weaknesses:**

- In the title, abstract, introduction, and related work, the claimed domain of this paper is too large. As this method is designed specifically for fluid dynamics, the title, abstract, introduction, and related work should reveal it. Otherwise, the readers will think that this method can be applied to the broad decision-making / RL scenarios.
- There are state-of-the-art control and simulation algorithms specifically designed for fluid control. The authors need to discuss them since they are highly relevant references. Also, these control methods should be the baselines to demonstrate the effectiveness of the paper.
- The demonstration of results in Section 4.1 and 4.3 is not suitable to be in the Methodology section, and should be in Experiment.
- Both the method and experiment are insufficient. The components of the algorithm are not novel, and most of them have been already proposed by existing works.

**Questions:**

- What is the relationship between $u_t$ in Eq.3 and $q$ in Eq.2? Also, there is $u$ in Eq. 1, but it demonstrates the velocity there.
- How to partition the region to get $D_c$ in Eq. 9?

---

### Official Review · Reviewer_XoiH · 2025-11-01

**Soundness:** 2
**Presentation:** 2
**Contribution:** 3
**Rating:** 4
**Confidence:** 2

**Summary:**

This work introduces PD-FS, a three-stage framework for reinforcement learning based control in fluid based environments. The authors argue that pure surrogate models for environment dynamics are fast but inaccurate, while classical CFD simulations are accurate but very inefficient in terms of runtime which can be prohibitive for training or exploration by the agent. So, they propose a sequential framework involving surrogate pertaining, CFD refinement, and PPO based real world transfer, aimed at unifying efficiency, fidelity and scalability. They test their framework on a fish locomotion task, and find that their framework works 50 times faster than full CFD simulation while having comparable fidelity.

**Strengths:**

1.	The authors leverage the structure of dynamics (i.e. dependance on tail-beat frequency) to tailor the surrogate model for increased fidelity while maintaining efficiency provided by data-driven methods in general, and their results show their success, at least in the domain considered. This is a proven strategy to improve model accuracy and stability.
2.	The efforts by the authors to study the sim to real gap by reproducing the fish locomotion experiment in hardware is commendable.

**Weaknesses:**

1. The authors, at multiple points in the paper, claim some property of their framework without providing sufficient (or any) evidence. First, they claim on line 72, that their model generalizes across regimes and perturbations, but their experiments are only limited to fish locomotion. The model or the learned policy is not demonstrated in other tasks. In this sense, the title of the paper is quite misleading, since the proposed methodology is only applicable to fish locomotion and not for general control. Second, around line 306, the authors claim that the learned policy exhibits robust propulsive stability and adaptive control capability, yet there are no experiments/ablations demonstrating that. In Appendix E, the authors attempt to demonstrate some ablations, but the results seem to be less numerical and more subjective, like ‘low’ or ‘high’ drift, ‘slow’ or ‘fast’ convergence speed, which is unsatisfactory.
2. The paper can be improved a lot writing and presentation wise. First, there are points like model drift from not using a partitioned frequency, or the three stage pipeline which is the main contribution, which are repeatedly mentioned unnecessarily. The authors can instead use the space to expand on some required background for a general reader, like the dependance of dynamics of locomotion on the tail-beat frequency which seems to be one of the central motivations of the network design. Second, most of  the figures and diagrams (both the ones explaining the methodology and the experiments) are not adequately explained, only mentioned briefly in the text. It would be very informative if you include this valuable information in the captions. For example, it is not clear at all what the swim modes represent in Figure 5.
3. In general, more experiments demonstrating the generalizability of the proposed methodology will greatly increase the chances of this work being accepted.

**Questions:**

1.	The authors mention that they use ‘lightweight’ surrogates in their framework. What do they mean by lightweight in this context?
2.	Does the kinematic state s_t include the fluid variables like velocity field as well? If not, how does the agent perceive the fluid environment?
3.	It might be good to include a figure or a diagram of the robotic fish hardware in the main paper, since that seems to be a significant contribution of this paper.
4.	How do you partition the dataset into the frequency partitions D_C in equation (9)?

---

### Official Review · Reviewer_wWYi · 2025-11-03

**Soundness:** 3
**Presentation:** 2
**Contribution:** 2
**Rating:** 4
**Confidence:** 3

**Summary:**

The paper proposes combining ML surrogates, high-fidelity simulations and real-world deployment in a staged fashion to speed up and stabilize the training of a reinforcement learning model to control a robotic fish swimming. They evaluate their model on a real world robotic fish platform as well as empircally showing the speed and stability gains when training RL with their 3 stage approach.

**Strengths:**

- Refining and testing their trained model on a robotic platform provides strong evidence their their learning method works in real world scenarios.
- The time tradeoff of using their 3 stage strategy versus a CFD only training is significant.

**Weaknesses:**

- While the ideas are generalizable, this method is hand tailored to the fish control scenario. For example, the mode classifier is designed for the 3 relevant modes in the fish experiment but other scenarios would require other bespoke design decisions.
- The authors state that they achieve "efficient, robust locomotion that generalizes across regimes and perturbations." As far as I can tell they only validate their model on the robotic platform on the "canonical task of straight-line swimming" which does not suggest generalization across different regimes and perturbations. I think to make this claim, the authors would have to experiment on either different tasks (swimming in straight lines, curved paths, circles, etc.) or under different environmental conditions (adding external forces to the water, etc.).

**Questions:**

- How much does the real world setting improve the performance of your framework?
- Do you only test/validate the model on the robotic platform? Or do you also train the RL with feedback from the this platform? For Table 2, Col 4 in the paper ("Experimental") you have a time cost of 10h, does that mean you trained the RL model with feedback from just the experimental robotic platform?
- How is the stability metrics $S_t$ computed?
- For the flow dynamics, are they modeled/simulated in 2D or 3D?
- How long does it take to train the data driven surrogate? How long does it take to generate the CFD data for the training of the surrogate model?

---

### Meta-Review · Area_Chair_inho · 2026-01-05

**Summary:**

The paper introduces a reinforcement learning–based framework for controlling the locomotion of a fish-like robot in a fluid environment. The authors first pretrain a lightweight surrogate model to approximate fluid–structure interactions between the robot and the surrounding flow. This surrogate is then fine-tuned using a more computationally demanding CFD solver, combined with a PPO algorithm operating in closed-loop control. The resulting controller is ultimately deployed on a real robotic platform. Experimental results demonstrate a favorable trade-off between control performance and computational cost, with a reported 50× speedup compared to a baseline relying exclusively on CFD-based training.
The reviewers appreciate the relevance of the problem, the overall strategy of leveraging a surrogate model to significantly reduce computational cost, and the successful deployment on a physical robotic platform, which strengthens the practical relevance of the work.

However, they identify several claims that are not sufficiently supported by the experimental evidence. First, the generality of the approach is questionable: while the high-level ideas could potentially extend to other control problems, the proposed framework appears highly specialized and tailored to the specific fish-like robot considered. Second, generalization across multiple regimes is not demonstrated, as the evaluation is limited to a single canonical task. Third, claims related to stability and adaptive control are not substantiated by dedicated experiments.

In addition, the reviewers raise concerns about the organization and clarity of the manuscript and suggest that further experimental validation would be necessary to fully support the paper’s claims.

**Reviewer Concerns:**

Although the authors revised the manuscript, they did not provide a rebuttal addressing the reviewers’ comments and questions.

**Reviewer Scores:**

No rebuttal

---

### Decision · Program_Chairs · 2026-01-26

Reject